# Heavy-to-light electron transition enabling real-time spectra detection of charged particles by a biocompatible semiconductor

**Dou Zhao[1,2], Ruiling Gao[3,4], Wei Cheng[5], Mengyao Wen[1], Xinlei Zhang[6], Tomoyuki Yokota [2], Paul Sellin [7], Shengyuan A. Yang [4], Li Shang [1], Chongjian Zhou [1], Takao Someya[2] ✉, Wanqi Jie[1] ✉ & Yadong Xu [1] ✉**

The current challenge of wearable/implantable personal dosimeters for medical diagnosis and radiotherapy applications is lack of suitable detector materials possessing both excellent detection performance and biocompatibility. Here, we report a solution-grown biocompatible organic single crystalline semiconductor (OSCS), 4-Hydroxyphenylacetic acid (4HPA), achieving real-time spectral detection of charged particles with single-particle sensitivity. Along in-plane direction, two-dimensional anisotropic 4HPA exhibits a large electron drift velocity of $5 \times 10^5$ cm s$^{-1}$ at "radiation-mode" while maintaining a high resistivity of $(1.28 \pm 0.003) \times 10^{12}$ Ω·cm at "dark-mode" due to influence of dense π-π overlaps and high-energy L1 level. Therefore, 4HPA detectors exhibit the record spectra detection of charged particles among their organic counterparts, with energy resolution of 36%, $(\mu t)_e$ of $(4.91 \pm 0.07) \times 10^{-5}$ cm$^2$ V$^{-1}$, and detection time down to 3 ms. These detectors also show high X-ray detection sensitivity of 16,612 μC Gy$_{abs}$$^{-1}$ cm$^{-3}$, detection of limit of 20 nGy$_{air}$ s$^{-1}$, and long-term stability after 690 Gy$_{air}$ irradiation.

Fast-response, biocompatible, and low-cost semiconductors are promising candidates as wearable/implantable dosimeters for real-time and position-sensitive radiation monitoring for people under radiation-exposure risks, e.g., astronauts, radiologists, and patients under radiotherapy. In particular, medical imaging diagnosis (e.g., computed tomography) and radiotherapy relying on ionizing radiations such as X/γ-rays and charged particles, are effective for the identification and cure of serious diseases such as cancers and COVID-19 pneumonia accounting for millions of deaths per year[1,2], while they also cause serious side-effects to humans organs, e.g., brain and heart, thus require real-time and position-sensitive radiation monitoring for safety control[3,4]. In addition, radiation monitoring can help

radiologists directly judge whether sufficient radiation has been absorbed by the tumor while simultaneously minimizing the radiation to surrounding healthy organs[5]. Currently, empirical in-vitro dose measurements are utilized to calculate the prescribed dose delivered to the tumor, but the accuracy is limited by the complex physical interaction between the beam source and tumor[6]. In addition, this method cannot recognize random or systematic errors once the dosimetry has been performed, such as sudden anatomical changes (e.g., colorectal gas, stomach filling), and systematic displacements that may affect the dose coverage of the target[7]. Therefore, wearable/implantable detectors for real-time, position-sensitive, and even in-vivo monitoring radiations absorbed by tumors and nearby healthy

[1]State Key Laboratory of Solidification Processing, Northwestern Polytechnical University, 710072 Xi'an, Shaanxi, China. [2]Department of Electrical Engineering and Information Systems, The University of Tokyo, Tokyo 113-8656, Japan. [3]International Center of Quantum and Molecular Structures, Shanghai University, 200444 Shanghai, China. [4]Research Laboratory for Quantum Materials, Singapore University of Technology and Design, Singapore 487372, Singapore. [5]Department of Nuclear Science and Engineering, Nanjing University of Aeronautics and Astronautics, 211106 Nanjing, China. [6]School of Physics and Information Technology, Shaanxi Normal University, 710119 Xi'an, Shaanxi, China. [7]Department of Physics, University of Surrey, Guildford GU2 7XH, UK. ✉e-mail: someya@ee.u-tokyo.ac.jp; jwq@nwpu.edu.cn; xyd220@nwpu.edu.cn

organs, are significant for optimizing the therapy plan or safety control, which requires detectors possessing high sensitivity, fast response, good biocompatibility, and low-cost.

A current challenge is the lack of suitable detector materials. The state-of-the-art radiation detectors based on inorganic semiconductors with single-photon sensitivity, e.g., CdZnTe, Si, and CsPbBr$_3$, either contain toxic metal elements to lose biocompatibility or have high fabrication costs due to the high-temperature melt process[8–12]. During the past decade, solution-grown organic single crystalline semiconductors (OSCS), e.g., 4-hydroxycyanobenzene (4HCB) and Rubrene, have been demonstrated as effective detectors for X-rays, charged particles, and neutrons[13–15]. Compared with the photocurrent detection mode commonly utilized by OSCS detectors, the spectral detection mode can probe both energy and count information with single-particle sensitivity, while the performance of OSCS is limited by the slow charge transport properties. For example, 4HCB OSCS exhibits superior spectra detection capabilities for charged particles, but it still requires a long duration time of around 30 min for a spectrum measurement, indicating a low detection efficiency thus limiting their real-time radiation detection applications[15].

The current reported organic detectors haven't achieved real-time spectroscopic detection for charged particles due to the commonly observed elongated rising time in a single electric pulse excited by one random charged particle[16]. The rising time approximates the charge-drift time between two electrodes, normally contains a fast-rising (below 1 μs) and accompanying slow-rising components (tens of microseconds) in organic detectors while that usually below 1 μs in inorganic detectors, e.g., CdZnTe[15,17]. Whether seeking novel materials or possible solutions to improve the charge transport in OSCS, the mechanism responsible for the two-stage rising pulse should be clarified, however, neither its origin nor the improved method is well understood[18,19].

4-Hydroxyphenylacetic acid (4HPA), an organic compound found in olive oil, is commonly used for drug synthesis, e.g., 3,4-dihydroxyphenylacetic acid, hasn't been investigated as a photodetector[20]. Here, we report that solution-grown 4HPA OSCS possessing superior electron transport properties achieves record spectra detection for charged particles with single-particle sensitivity. The superior performance originates from a radiation-stimulated heavy-to-light electron transition, which is clarified here for the first time by combining experiments and calculations. In particular, the major anisotropy of 4HPA OSCS in terms of the two-dimensional (2D) crystallographic structure and the corresponding charge transport properties are compared along in-plane (a axis) and out-of-plane (c axis) directions. Utilizing Density Functional Theory (DFT) calculations and Monte Carlo (MC) simulations, a radiation-stimulated heavy-to-light electron transition is predicted, which is consistent well with the charge transport characterization using the time-of-flight (TOF) method. In addition, the microstructural origin for exciton dissociation is also clarified using photocurrent data and atomic force microscope (AFM). Finally, real-time spectral detection of charged particles and other ionizing radiations such as X-rays and neutrons are investigated using our 4HPA OSCS detectors with proper crystallographic orientation.

## Results and discussion
### Chemical composition and crystallographic structure of 4HPA OSCS
The physical properties of OSCS are determined by their chemical composition and crystallographic structures. In the 4HPA molecule (Fig. 1a), two opposite hydroxyl groups (-OH) and acetoxy group (-CH$_2$COOH) on benzene leads to 2D crystallographic structures and planar appearance of 4HPA OSCS (Fig. 1b) obtained from the simple solution method (Fig. S1). Due to the metal-free compositions, 4HPA OSCS has excellent biocompatibility with cell viability over 90% after 24-h incubation with a 4HPA concentration of 2 mg ml$^{-1}$ (Fig. 1c and

Fig. S2) is much better than 4HCB OSCS (Fig. S3), making 4HPA promise as implantable/wearable devices for healthcare monitoring applications. For example, we can fabricate a 4HPA thick film sensor with a thickness of around 100 μm as we demonstrated for 4HCB[21]. With good radiation detection performances and flexibility, such radiation sensors could be integrated with organic photovoltaic modules such as power supply[22], small customized application-specific integrated circuit (ASIC) chips for data collection and processing[7], and wireless communication modules. The whole device could be directly attached to human skin due to the biocompatibility of 4HPA, to achieve a lightweight, flexible, and comfortable radiation monitoring sensor. Then the radiation monitoring signal could be uploaded to a cell phone or personal computer, achieving self-powered and wireless radiation monitoring.

The 4HPA OSCS (Fig. 1d) is the orthorhombic system with space group of P2$_1$2$_1$2$_1$ (No. 19), the unit cell a = 0.5 nm, b = 0.9 nm, and c = 1.5 nm. This structure enables 4HPA OSCS to possess a high degree of π–π overlap along the a axis (in-plane direction) while intermolecular hydrogen bonds (-O$_I$H...O$_{II}$-, -O$_I$H...O$_{III}$-) along the c axis (out-of-plane direction) (Figs. S4 and S5). The powder- (Fig. S6) and single-crystal (Fig. 1e) XRD patterns demonstrate that the largest exposure facet is (001). Thermal stability is a common issue for practical applications of organic radiation detectors. The DSC-TG result shows that 4HPA OSCS has excellent stability below its melting point (146 °C) (Fig. 1f), which is better than 4HCB (melting point of 123 °C)[23].

### Band structure of 4HPA OSCS
The periodical molecular packing and weak intermolecular forces lead to a discontinuous band structure and a quasi-band charge transport in 4HPA OSCS, which is revealed by DFT calculations (Fig. 2a). The space between the highest occupied and the lowest unoccupied molecular orbitals (HOMO and LUMO levels) is the forbidden band. The theoretical indirect bandgap energy is 3.75 eV, very close to the experimental bandgap of 3.97 eV by a Tauc plot fit of the UV–Vis transmittance spectrum (Figs. S7 and S8). The wide bandgap prohibits the sensitivity of the 4HPA detector to visible light, benefiting a low background current.

The LUMO level of 4HPA OSCS is flatter than the L1 level with higher energy (Fig. S9), indicating the varying effective masses of electrons. Corresponding to the 2D structure of 4HPA, the anisotropic effective mass m* of L1, LUMO, HOMO, and H1 levels along both a and c axes is calculated by DFT, with 1/m* values shown in Fig. 2b and Table 1. In general, the a axis of 4HPA OSCS has a larger 1/m* than the c axis. In addition, the LUMO level has a smaller 1/m* than the HOMO level while the L1 level has a significantly larger 1/m* than LUMO and H1 levels. Figure 2c, d gives the electron cloud density of LUMO and L1 levels along a and c axes, respectively, which are consistent with the intermolecular π–π overlaps along the a axis and hydrogen bonds along the c axis. According to $\mu = \frac{q\tau}{m^*}$ (q and τ are elementary charge and mean free time of electrons/holes, respectively), electrons at the L1 level along the a axis, theoretically possess a larger charge mobility (with 1/m* = 4.98) than electrons at other levels (H1, HOMO and LUMO levels).

### Charge transport behaviors of 4HPA OSCS under radiation stimulation
The band structure of 4HPA indicates an unusual charge transport mechanism under radiation, which is investigated by the TOF method with α particle irradiation, as shown in Fig. 3a[24]. Contrasting transport properties along the a and c axes of 4HPA OSCS are measured using detectors with sandwich and coplanar electrode configurations (Fig. 3b), respectively. The shallow penetration depth (35 μm, Fig. 3c and Figs. S10–S11) of α particles makes electron-only and hole-only transient charge-drift pulses can be measured separately. For example, when an alpha particle is incident on the cathode, electrons drift across the detector whilst holes get recombination at the cathode, generating

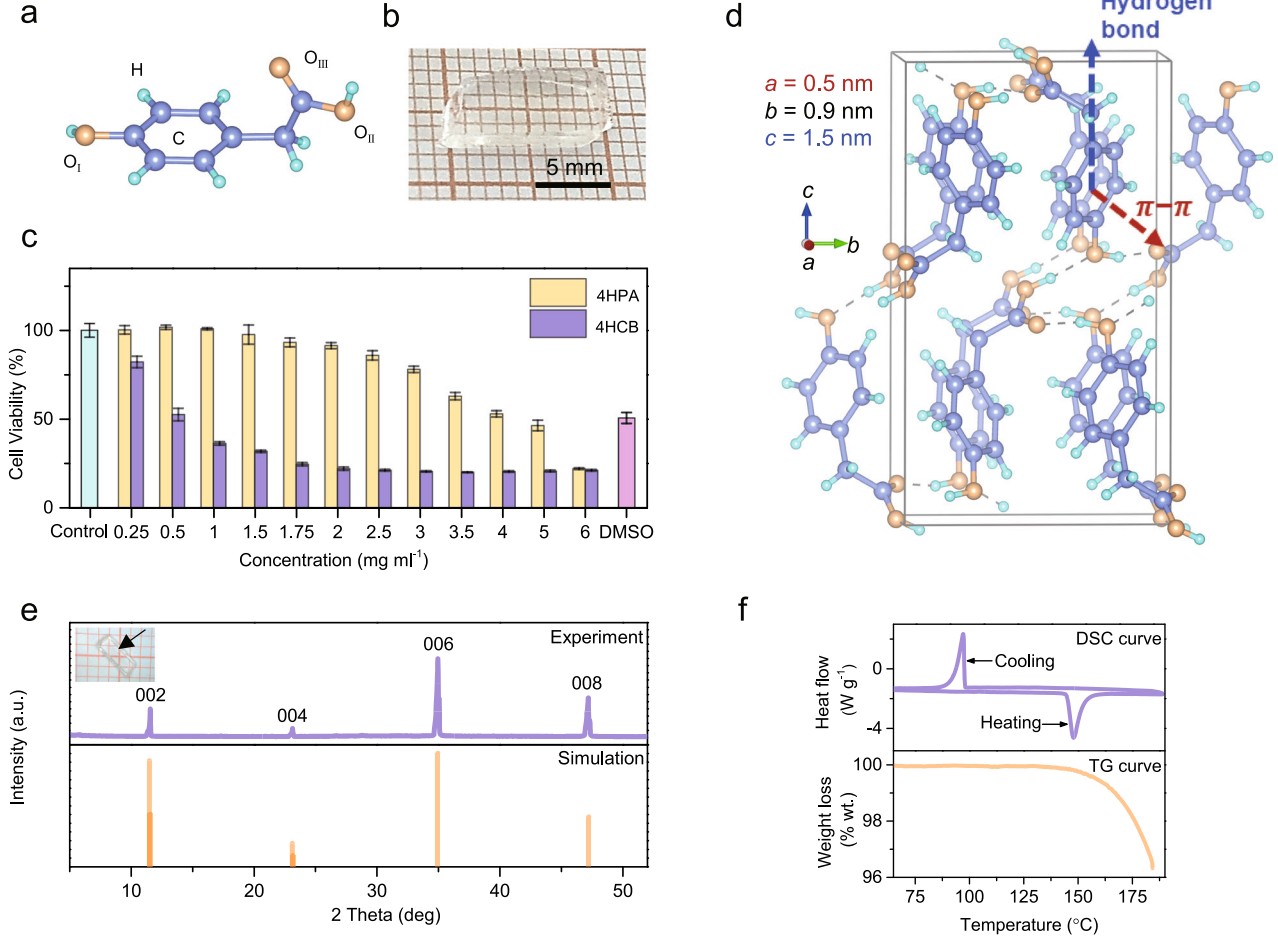

**Fig. 1 | Chemical and crystallographic structures of 4HPA OSCS. a** Molecular formula of 4HPA, $O_I$, $O_{II}$, and $O_{III}$ representing different O occupations, **b** typical photograph of centimeter-size 4HPA single crystals with *a*, *b*, and *c* means the lattice parameter of main axes of 4HPA unit cell, **c** cytotoxic responses of HUVECs cells to 4HPA compared with 4HCB, **d** 4HPA OCSC molecular packing in 1 × 1 × 1 unit cell viewing from the *a* axis, **e** single-crystal XRD pattern of the largest exposure (001) facet, **f** DSC-TG curve of 4HPA OSCS.

an electron-only rising pulse. In addition, Fig. 3d, e gives the Monte Carlo simulated energy spectra of α particles excited electrons after first ionization and multiple ionization, respectively, indicating that radiation-generated electrons possess energy of 0.8–150 eV before the thermal relaxation.

Figure 3f shows typical rising pulses averaged by 200 electron-only pulses along the *a* axis at electric fields from 800–4000 V·cm⁻¹, each of which consists of fast-rising and remarkably slow-rising components that are commonly observed in organic semiconductors. Generally, electrons along the *a* axis have a faster drift time than the *c* axis (Fig. 3g). It is worth noting that electrons drift faster than holes along both *a* and *c* axes at same electric fields (Figs. S13 and S14), which is contradictory with theoretical 1/*m*\* values of HOMO level (e.g., 3.06 along the *a* axis) and LUMO level (1.38) but is consistent with the theoretical 1/*m*\* of H1 level (3.79) and L1 level (4.98). This suggests that macroscopic charge transport behaviors may be affected by the L1 level, or the ionized electrons may also have a certain proportion on the L1 level even after the thermal relaxation process when two conditions that the high-energy radiations are continuously injected into 4HPA detectors and a high electric field is applied are satisfied at the same time. Then fast-rising and slow-rising stages in the rising pulse may originate from hybrid influence from L1 and LUMO levels, respectively.

In addition, the saturated amplitude of hole-only and electron-only pulses is anisotropic (Figs. S13 and S14). Along the *a* axis with the electric field of 2800 V cm⁻¹ (Fig. 3g, h), the saturated amplitude of

hole-only pulses is only 0.7 mV, an order magnitude lower than the electron-only pulse (10 mV). This contrasts with the *c* axis where amplitudes of both hole-only and electron-only signals are around 10 mV at 2800 V cm⁻¹. According to TOF theory[25], the saturated amplitude is approximately positive to μ*T* (SI 1, μ is carrier mobility, *T* is average drift time of electrons or holes before being trapped), which indicates the μ*T* of holes along the *a* axis, $\mu_{h\text{-}a}T_{h\text{-}a}$, is one order smaller than electrons $\mu_{e\text{-}a}T_{e\text{-}a}$ and the μ*T* products along the *c* axis, $\mu_{h\text{-}c}T_{h\text{-}c}$, $\mu_{h\text{-}e}T_{h\text{-}e}$ when the electric field is fixed. According to Table 1, the theoretic $\mu_{h\text{-}a}$ is close to $\mu_{e\text{-}a}$, larger than $\mu_{h\text{-}c}$ and $\mu_{e\text{-}c}$, which indicates that $T_{e\text{-}a}$ may be one order of magnitude smaller than $T_{e\text{-}a}$, $T_{h\text{-}c}$, and $T_{e\text{-}c}$, and a significant hole trapping may occur along the *a* axis.

According to the above results, α particles excited electrons have higher mobility and less trapping possibility than holes, dominating the best charge transport behaviors in 4HPA. In the absence of significant charge trapping, the electron mobility can be calculated by[25]:

$$\mu = \frac{v_{dr}}{E} = \frac{d}{E \times t_f} \quad (1)$$

where $v_{dr}$ is drift velocity, *E* is electric field, *d* is distance between two electrodes, and $t_f$ is carrier drift time which is given by the 10%–90% rise time of the charge pulse.

The electron-only drift velocity ($v_{dr}$) versus the electric field (*E*) relations shown in Fig. 3i are nonlinear for both *a* and *c* axes, leading to the field-dependent electron mobility. For example, along the *a* axis,

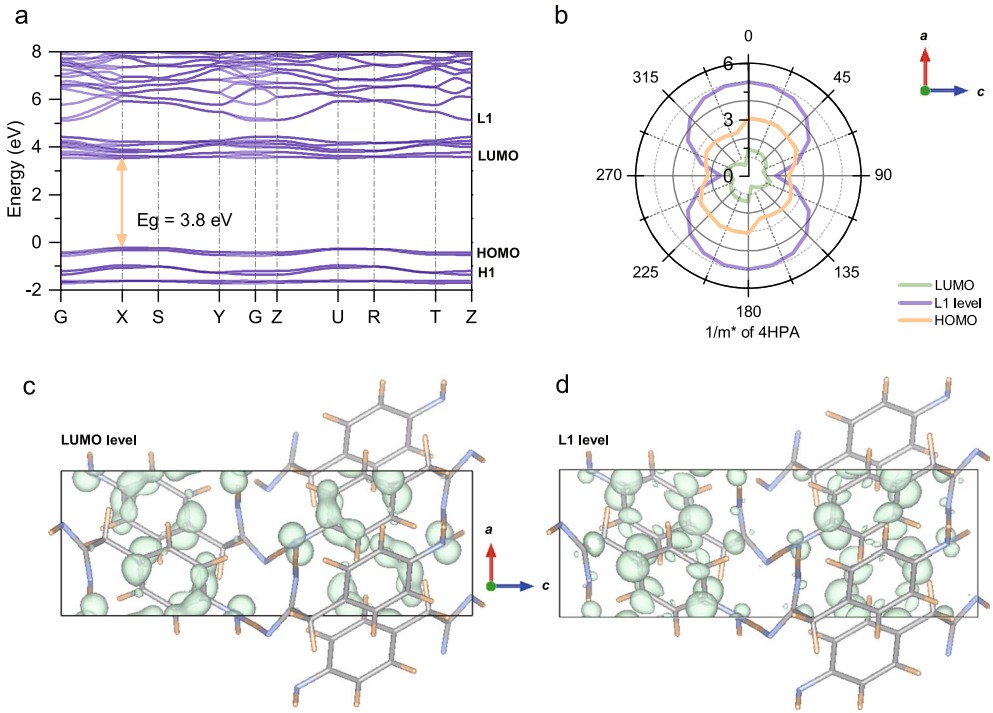

**Fig. 2 | Band structure of 4HPA OSCS. a** The energy band structure, $E_g$ is the bandgap of 4HPA OSCS, **b** anisotropic $1/m^*$ ($m^*$ is the effective mass) along $a$ and $c$ axes of 4HPA OSCS, and crystal structures viewing along $b$ axis and corresponding electron cloud density (green clusters) of **c** LUMO, **d** L1 levels.

the electron mobility is $0.84 \pm 0.01\ \text{cm}^2\,\text{V}^{-1}\,\text{s}^{-1}$ ($\mu_{a1}$) when the $E$ is under $2800\ \text{V}\,\text{cm}^{-1}$ (the 'critical' electric field value ($E_c$)), then increases to $4.17 \pm 0.04\ \text{cm}^2\,\text{V}^{-1}\,\text{s}^{-1}$ ($\mu_{a2}$) above $E_c$. Simultaneously, the electron-only pulse containing only a fast-rising component with a short drift time (e.g., $0.52\ \mu\text{s}$ at $2800\ \text{V}\,\text{cm}^{-1}$ along the $a$ axis (Fig. 3j) starts to occur above the $E_c$ for both axes, and the occurrence increases at higher electric fields.

The nonlinear relationship between $v_{dr}$ and $E$ is partly related to the fast-rising component and remarkably slower-rising component in α-particle induced charge pulses. However, the experimentally observed relative mobility ratios of "fast" electrons and "slow" electrons ($\delta = \frac{\mu_{f-e}}{\mu_{s-e}}$, 4.96 for $a$ axis and 17.33 for $c$ axis) in Fig. 3i are different from those obtained by theoretical calculation (3.61 for $a$ axis and 1.60 for $c$ axis in Fig. 2b), which may originate from the standard charge transport model. In Eq. (1), the excited electron-hole pairs are assumed to drift apart and dissociate instantaneously after their creation[26]. This is reasonable in inorganic detectors exciton dissociation behavior is neglected while the exciton behavior plays a significant role in organic detectors[27]. The excitons diffusion and dissociation processes will elongate macroscopic drift time, contributing to the slow mobility $\mu_{s-e}$ in organic detectors. Therefore, a correction factor of ($f(p)$) that represents exciton dissociation possibility, describes the difference between the experimental $\mu_{s-e}(ex)$ and the theoretical $\mu_{s-e}(th)$ is introduced in TOF theory for experimental charge mobility:

$$\mu_{s-e}(ex) = f(p) \times \mu_{s-e}(th) \quad (2)$$

Then for the electron mobility, the corrected expression is:

$$\delta(ex) = \frac{\mu_{f-e}(th)}{f(p) \times \mu_{s-e}(th)} = \frac{\delta(th)}{f(p)} \quad (3)$$

The calculated $f(p)$ along the $a$ and $c$ axis are 0.73 and 0.09, respectively, which indicates a higher exciton dissociation possibility along the $a$ axis of 4HPA.

## Surface-edge-states assisted exciton dissociation along in-plane direction

The consideration of exciton can explain the significant slow-rising component of the pulse in Fig. 3, which influences the macroscopic transport properties of 4HPA detectors. However, two issues how excitons influence the transport in 4HPA detectors and why the $a$ axis of 4HPA has a higher exciton dissociation possibility than the $c$ axis remain unclear. In organic semiconductors, either the donor/acceptor interface or the high electric field is required for exciton dissociation[28]. As-grown 4HPA single crystals contain imperfections, some of which can act as donor/acceptor interfaces to promote exciton dissociation with the assistance of an electric field. Under a low electric field, the exciton dissociation is limited to the concentration of the donor/acceptor interfaces, inducing the excitons-dissociation-dominated transport behaviors (corresponding to remarkably smaller charge mobility). The high electric field can efficiently dissociate excitons thus producing the charge-drift dominated transport behaviors (the higher charge mobility).

In addition, layer edges in organic detectors with positive/negative electric potentials can act as donor/acceptor interfaces[29,30]. The dense surface layer edges of the thickness of 1.5 nm (Fig. 4a) are formed due to the quasi-2D crystallographic structure and corresponding self-assembly growth of 4HPA OSCS from the solution method, which promotes exciton dissociation along the in-plane direction ($a$ axis) rather than the out-of-plane ($c$ axis). Along the $a$ axis, the significant hole-only trapping effect is deduced in Eq. (3), which is substantially absent along the $c$ axis. That indicates these surface layer edges may trap holes to promote exciton dissociation.

**Table 1 | Anisotropic $1/m^*$ value along the $a$ and $c$ axes of 4HPA**

| $1/m^*$ | HOMO | H1 level | LUMO | L1 level |
|---|---|---|---|---|
| $a$ axis | 3.06 | 3.79 | 1.38 | 4.98 |
| $c$ axis | 2.14 | 0 | 0.92 | 1.47 |

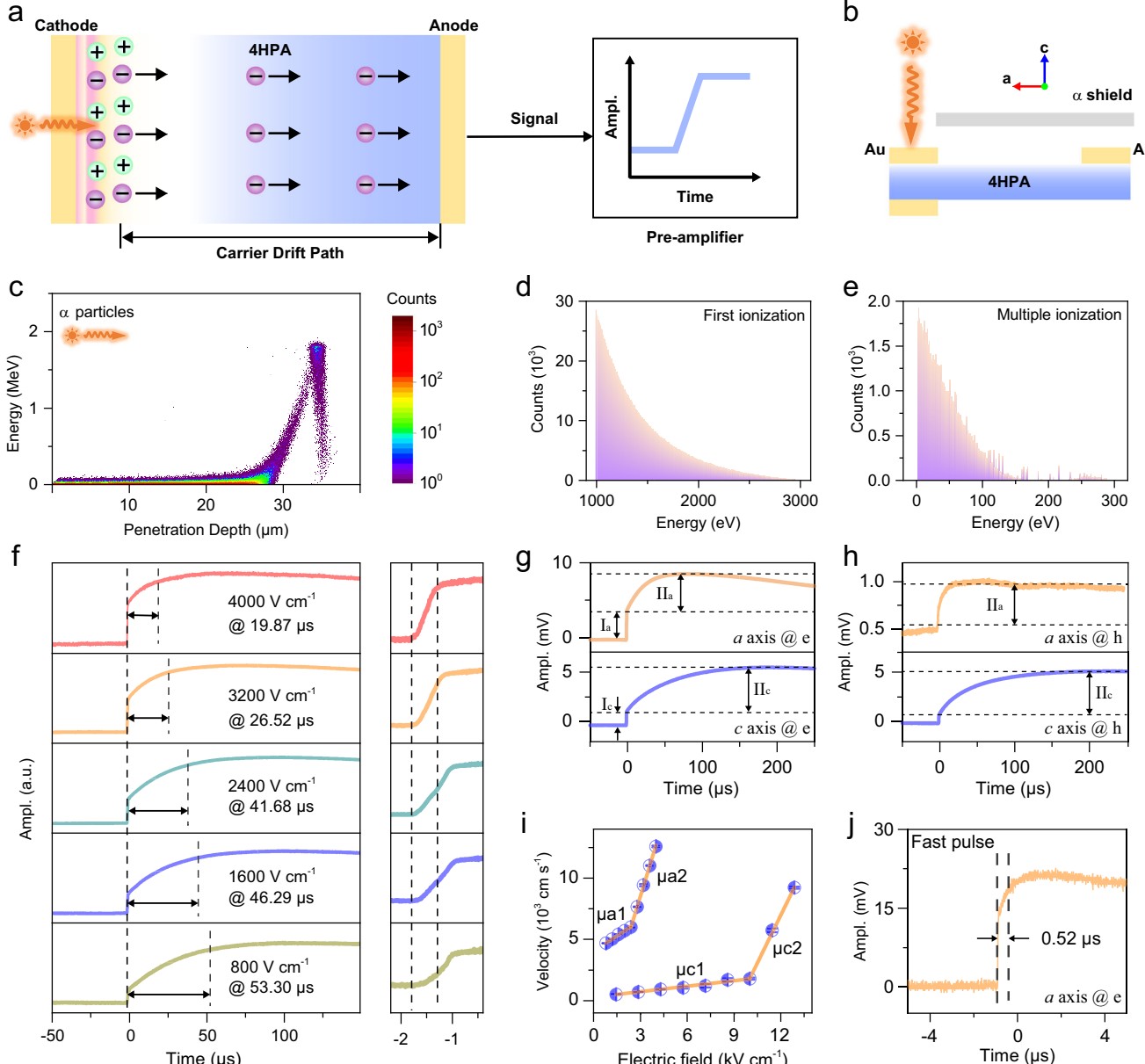

**Fig. 3 | Anisotropic electric properties of 4HPA single crystals. a** schematic diagram of α-TOF method for measuring electron-only transient pulse, **b** architectures of sandwich and coplanar devices for measuring electric properties along *a* and *c* axes, respectively, **c** energy deposition of 5.49 MeV α particles in 4HPA detectors changes with detector thickness, energy distribution spectra of electrons ionized by 5.49 MeV α particles of **d** first ionization and **e** multiple ionization, **f** (from down to up) time-resolved average electron-only drift pulses from α-TOF method measurement along *a* axis of 4HPA with bias voltages from 800 V cm⁻¹ to 4000 V cm⁻¹, **g** average electron-only drift time along *a* and *c* axes under the bias of 2800 V cm⁻¹, **h** average hole-only drift time along *a* and *c* axes under the bias of 2800 V cm⁻¹, **i** anisotropic electron drift velocity changes with electric fields along *a* and *c* axes, **j** single electron-only pulse with a rising time of 0.52 μs with a bias voltage of 2800 V cm⁻¹ along *a* axis.

Therefore, a low electric field (~2800 V cm⁻¹) is required for efficient exciton dissociation along the *a* axis, due to layer-edge assisted exciton dissociation (LAED) and electric-field assisted exciton dissociation (EAED) shown in Fig. 4b. However, the influence of nanoscale surface layer edges along the *c* axis is limited because excitons can diffuse to the bulk region, thus EAED dominates the excitons dissociation, and a higher critical field (~11000 V cm⁻¹) is required (Fig. 4c). This model is consistent with the higher exciton dissociation possibility (73%) along the *a* axis, compared to 9% along the *c* axis, as calculated by Eq. (3).

X-ray-induced photocurrents are then characterized to prove the carrier trapping effect of surface layer-edge states. Along the *a* axis, background current (around 12 pA at an electric field of 100 V cm⁻¹,

bulk resistivity of $(1.28 \pm 0.002) \times 10^{12} \, \Omega$ cm) is induced by free electrons, while the same number of holes are trapped by layers edge states (Fig. 4d). Under low-intensity illumination (L1-light in Fig. 4e), trapped holes absorb photons for de-trapping (DT process), then recombine (RE) with free electrons, resulting in the decreased photocurrent. With higher intensity illumination (L2-light in Fig. 4e), new electron-hole pairs are generated, resulting in increased photocurrent. Due to the limited influence of surface layer-edge states within 4HPA detectors along the *c* axis, the low-intensity illumination can generate new electron-hole pairs within the bulk region and increase the photocurrent (Fig. 4f). Figure 4g shows experimental X-ray photocurrent data acquired along the *a* and *c* axis, in which the X-ray intensity is controlled by the tube current of 5–50 μA. The photocurrent along the

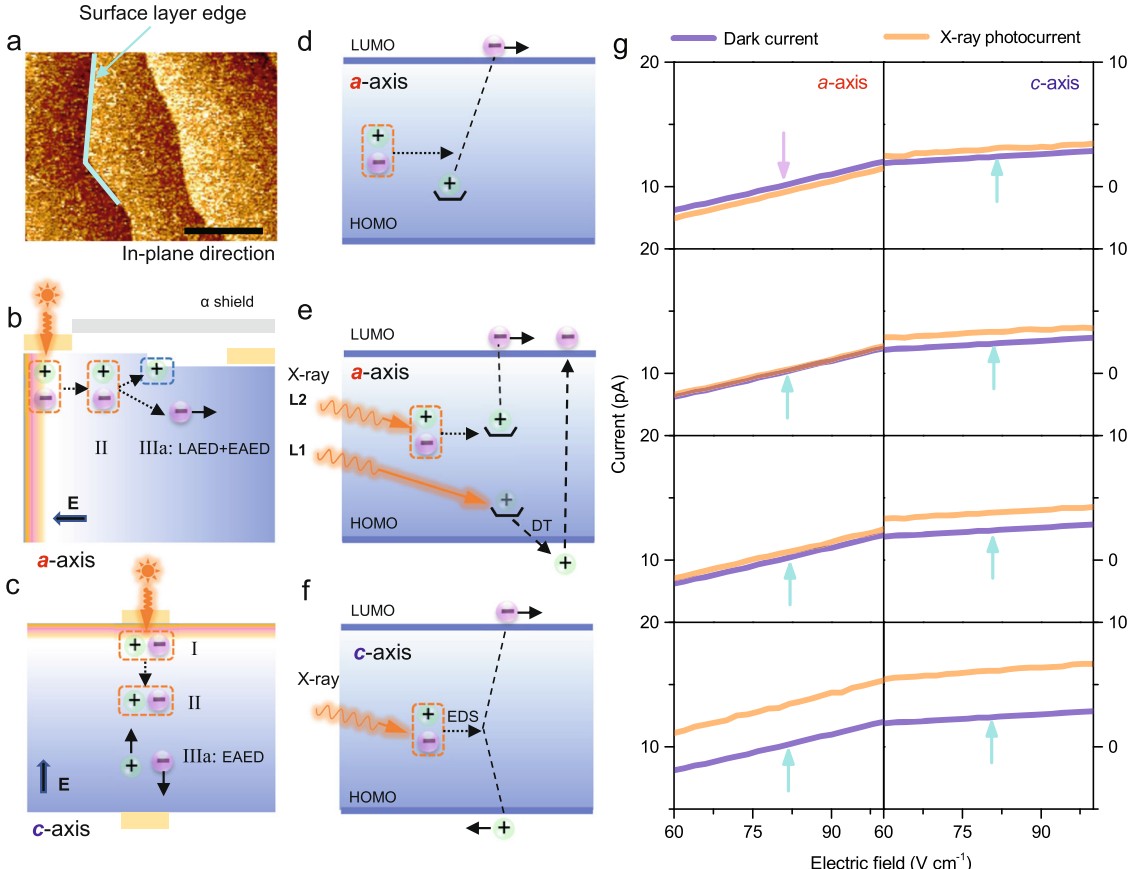

**Fig. 4 | Surface layer edge with positive potential to promote exciton dissociation along *a* axis. a** AFM image of surface layer-edges with a layer thickness of 1.5 nm (black scale bar: 150 nm), schematic diagram of layer edges promoted excitons dissociation along **b** *a* axis and **c** *c* axis (*E* is the electric field, LAED and EAED are Layer-Edge Assisted Exciton Dissociation and Electric-Field Assisted Exciton Dissociation processes, respectively), schematic diagram of surface layer-edge states in the energy band of 4HPA single crystals and its roles for **d** excitons dissociation along axis under dark condition, and anisotropic photocurrent along **e** *a* axis and **f** *c* axis (L1 and L2 are increased intensity of incident X-ray, RE, DT, and EDS are recombination, de-trap, and exciton dissociation processes, respectively), and **g** corresponding experimental results of anisotropic photocurrent of 4HPA single-crystal detectors along the *a* axis and the *c* axis (from left to right increase the power of incident X-ray beams) to prove the role of surface layer-edge states.

*a* axis is lower than the background current when the X-ray tube current is below 10 μA, while the photocurrent along the *c* axis is always higher than the background current, indicating the significant trapping effect that is present along the *a* axis.

According to the discussion in Figs. 3 and 4, the mechanism accounting for the fast-rising and slow-rising pulse components in α particle TOF measurements has been clarified. α particles produced electrons may be affected by the L1 level, resulting in "heavy-to-light electron transition" macroscopic transport behaviors, determining how fast the electron can transport under radiation. In addition, the significant slow-rising component is attributed mainly to the exciton dissociation process, which can be promoted by surface layer edges along the *a* axis. Therefore, efficient electron transport and exciton dissociation occur simultaneously along the *a* axis of 4HPA, which is a major factor in determining the good pulse-mode performance for ionizing radiation detection.

### Real-time charged-particle spectra detection by biocompatible 4HPA

The spectral detection performance of 4HPA detectors is evaluated under irradiation of 5.49 MeV α particles. It is worth noting that the large difference in electron and hole mobility will not influence the spectra detection of alpha particles if alpha particles are irradiated from the cathode side and the ionized electrons are collected from the anode. Figure 5a is the measured spectra (electrons-only signal) by

coplanar 4HPA detectors (*E//a* axis) at a series of bias voltages, with the best energy resolution of around 36% among organic detectors. Although the energy resolution of 4HPA detectors is still not very good when compared with the state-of-the-art inorganic detectors, e.g., CdZnTe and Si, there is plenty of space to improve the energy resolution of such organic detectors by various molecular engineering, optimization of the materials purity, crystallization quality, device structure, and measurement electronics.

The corresponding peak center versus bias voltage relationship is plotted in Fig. 5b, by which the electron's mobility-lifetime product can be extracted using the Hecht function[16]:

$$\eta = \frac{\mu\tau V}{d^2}\left(1 - \exp\left(-\frac{d^2}{\mu\tau V}\right)\right) \qquad (4)$$

where *V* is the applied voltage, $\mu\tau$ is the mobility-lifetime product. The fitting of the experimental data in Fig. 5b extracts the $(\mu\tau)_e$ product of $(4.91 \pm 0.07) \times 10^{-5}$ cm$^2$ V$^{-1}$ (along *a* axis). Figure 5c shows the spectra result (electron-only signal) based on sandwich 4HPA detectors (*E//c* axis) at bias voltages in the range of 400–1000 V. Figure 5d gives the corresponding Hecht function fitting of $(\mu\tau)_e$ along the *c* axis, which is $(4.62 \pm 0.08) \times 10^{-6}$ cm$^2$ V$^{-1}$. Fig. S16 shows hole-only spectra that lose energy discrimination ability. The higher $(\mu\tau)_e$ along the *a* axis originates from both the higher electron mobility and the prolonged electron lifetime by hole trapping at surface layer edges. As a result,

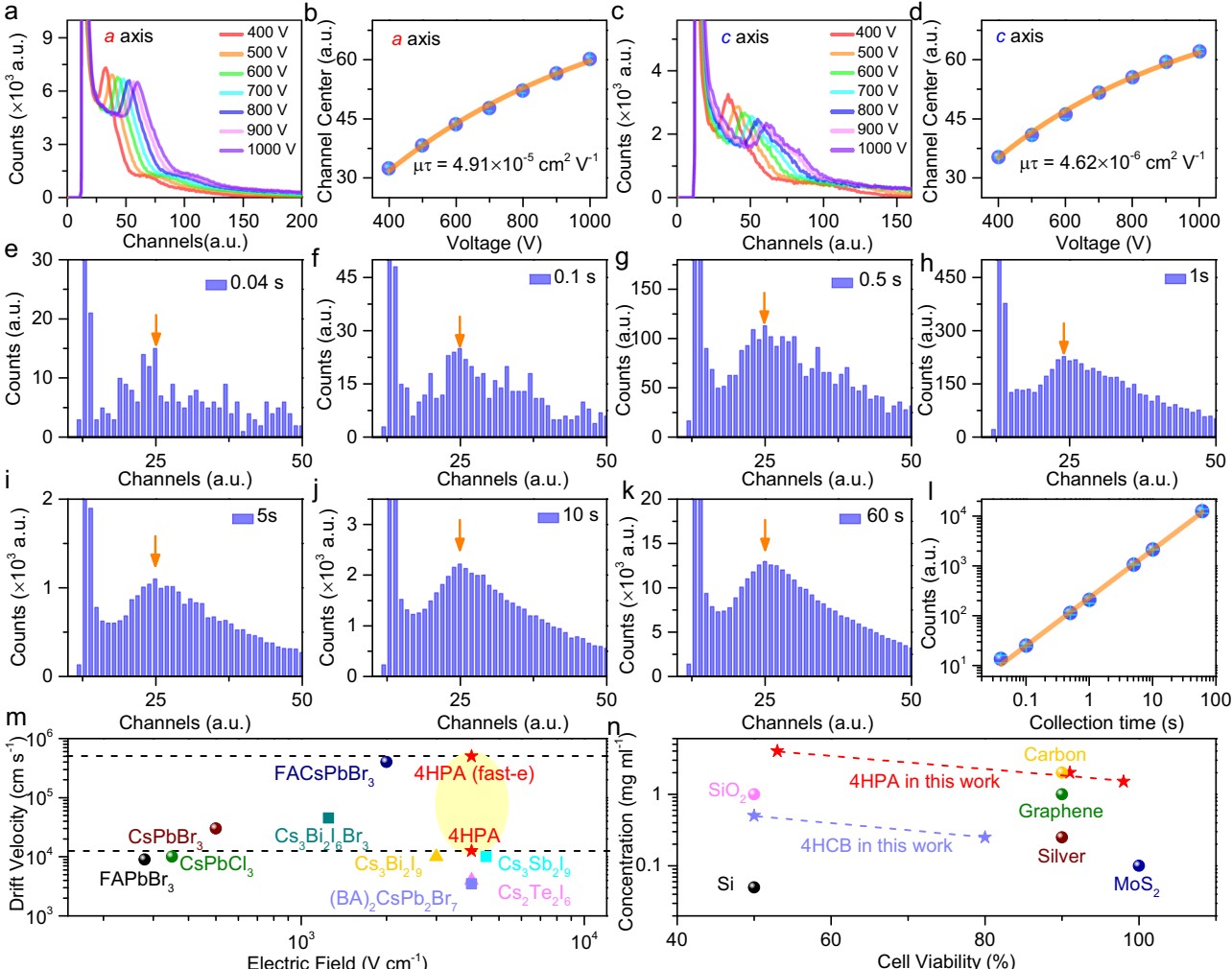

**Fig. 5 | Real-time charged-particle spectra obtained by electrons-only signals in 4HPA detectors. a** Spectra at a series of bias voltages and **b** channel centers vs. bias voltage curve for Hecht fitting along *a* axis ($\mu\tau$ is mobility-lifetime product), **c** spectra and **d** channel centers vs. bias voltage curve along *c* axis, **e–k** charged-particles spectra with detection time from 0.04 s to 60 s, **l** linear relationship between measured counts and incident charged-particles dose, **m** drift velocity comparison between 4HPA in this work and the state-of-the-art perovskite charged-particle's detectors[31–41], **n** biocompatibility comparison between 4HPA in this work and other materials[42–46].

the α particle spectra along *a* axis have superior energy discrimination and count abilities.

Efficient detection performance along the *a* axis promises a low operating voltage, and most importantly, pulse-mode operation at higher event rates compared with other organic detectors, leading to the real-time beam monitoring using a biocompatible 4HPA detector that has not been achieved with existing organic semiconductors. Figure 5e–k gives the alpha particle spectra with detection time from 0.04 s to 60 s at a bias voltage of 200 V, the incident alpha particle activity (**A**) in the order of $10^6$ Bq, which is around 10 times higher than alpha source for 4HCB[23]. The peak centers (corresponding to the energy of incident alpha particles beam) of spectra with different detection times are constant (No. 25 channels), and the amplitude of the peak centroid shows a linear relation with detection time (proportional to the beam dose), as shown in Fig. 5l. In this geometric setup, a pulse-mode acquisition time of 3 ms is sufficient to acquire a resolvable spectrum, with a peak centroid intensity of 10 counts. Thanks to the tolerance to high electric field and fast-electron transport at "radiation-mode", 4HPA detectors exhibit a usual large drift velocity of up to $5 \times 10^5$ cm s$^{-1}$ (SI 2), even higher than the superior halide perovskite materials (Fig. 5m)[31–41] while it has the high resistivity of $(1.28 \pm 0.002) \times 10^{12}$ Ω cm at "dark mode". In addition, 4HPA OSCS

also shows very good biocompatibility as carbon-based materials Fig. 5n)[42–46]. To our knowledge, there is no biocompatible organic semiconductor detector that can achieve real-time spectra detection for charged particles with single-particle sensitivity. In addition, the 4HPA detectors have a response for fast neutrons (Figs. S17 and S18).

## X-ray detection and imaging by 4HPA detectors using integration mode

For the clinical application, real-time and high-sensitive X-ray detection with integration mode is also important, which is demonstrated by 4HPA detectors along the *a* axis.

Figure 6a shows the typical photocurrent-time (*I–t*) curve of 4HPA detectors, with a dose rate range of 0.35–2.34 μGy$_{air}$ s$^{-1}$. The bias voltages of 5–130 V are applied along the *a* axis of 4HPA single crystals. No hysteresis or appreciable current drift is observed, which is comparable with other high-performance detectors[47–51]. The dark current is stable after repeated exposure under X-ray beam, with a value of 0.1 pA at a bias voltage of 150 V and a very low dark current drift -10$^{-9}$ nA cm$^{-1}$ s$^{-1}$ V$^{-1}$ (Fig. S19). The stable and repeatable detectable dose rate 0.35 μGy$_{air}$ s$^{-1}$ has been experimentally achieved (Fig. S20). The signal-to-noise ratio (SNR) of photocurrent varying dose rate at a series of bias voltage were calculated (SI 3), by which the detection of

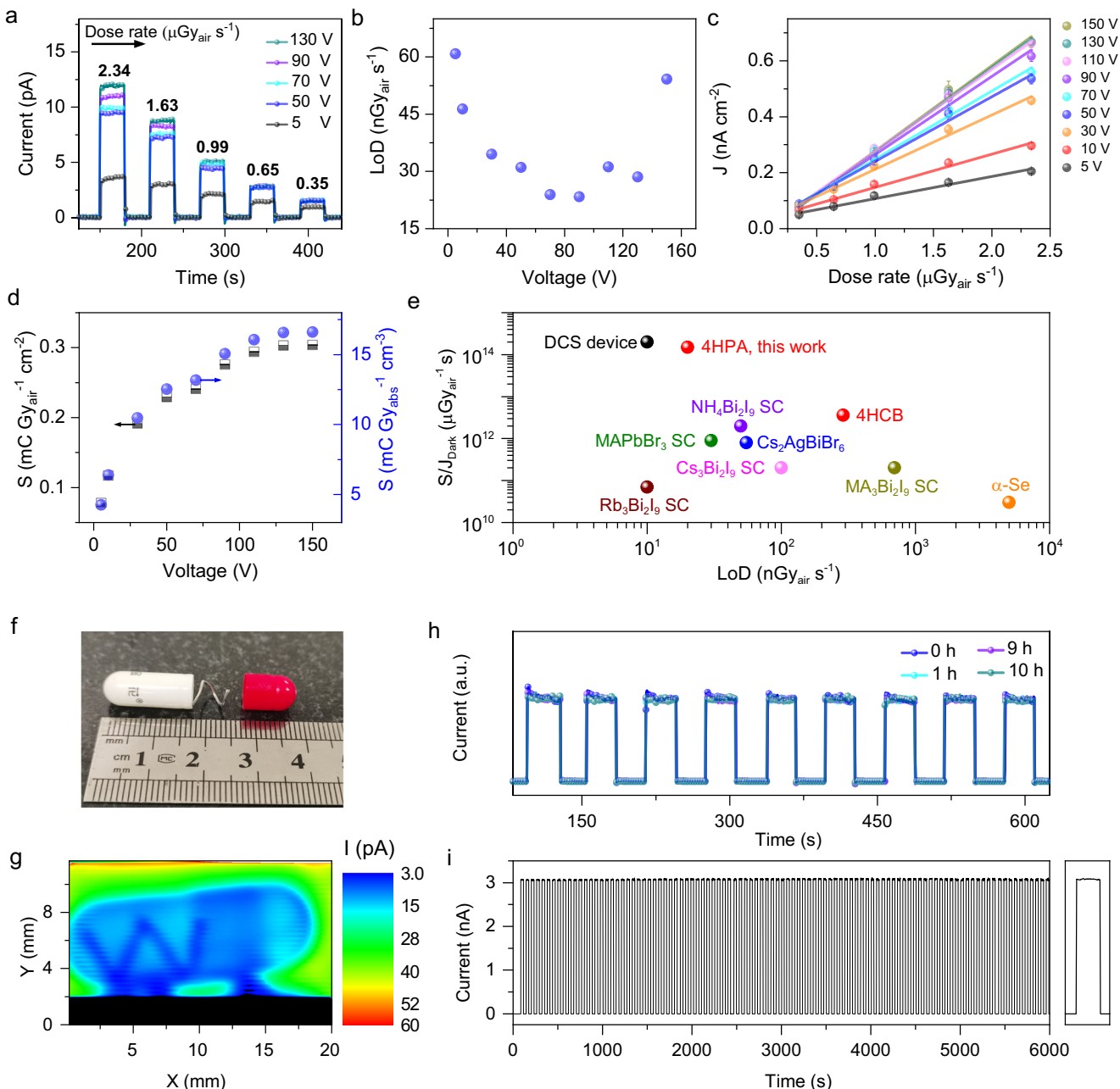

**Fig. 6 | X-ray detection and imaging performance of 4HPA detectors. a** *I*–*t* curves with decreased X-ray dose rate and switching X-ray "ON/OFF" at bias voltage from 5–130 V, **b** LoD changes with bias voltage, **c** photocurrent density (*J*) changes with irradiated X-ray dose rate at bias voltage from 5–150 V, **d** Sensitivity (*S*) of 4HPA detectors for 40 kVp X-ray at a series of bias voltage, **e** detection performance comparison of 4HPA and other organic and halide perovskite X-ray detectors in terms of the *S*/*J*$_{Dark}$ and the detection of limit (LoD) values, **f** metal wire in a plastic capsule and **g** its X-ray imaging by 4HPA imager detectors, **h** continuous 10-hour X-ray irradiation (150 kVp, total dose 690.48 Gy$_{air}$) stability, and **i** long-term work stability.

limit (LoD) values at each bias voltage (Fig. 6b) were estimated according to the SNR value larger than IUPAC[52] defined 3. The best LoD of 20 nGy$_{air}$ s$^{-1}$ is 275 times lower than the required 5.5 μGy$_{air}$ s$^{-1}$ for regular medical diagnostics[53].

Then the linear Photocurrent density and Dose rate (***J-D***) relation was shown in Fig. 6c, by which the Sensitivity (***S***) was calculated using linear fitting, with ***S*** of 330 μC Gy$_{air}^{-1}$ cm$^{-2}$ at a bias voltage of 150 V or 16,612 μC Gy$_{abs}^{-1}$ cm$^{-3}$ when considered the X-ray absorption of 4HPA detectors (Fig. 6d and SI 4). Then, the X-ray detection sensitivity (***S***)/Dark current density (***J***$_{Dark}$)[54] as high as 1.5 × 10$^{14}$ μGy$_{air}^{-1}$ s was achieved by 4HPA detectors for 40 kVp X-ray detection (SI 5). In general, 4HPA shows superior X-ray detection performances using the integrated mode, which is even comparable with the state-of-the-art

halide perovskite X-ray detectors (Fig. 6e)[54–62]. The good detection performance can be achieved in 4HPA organic detectors with weak X-ray absorption rate (SI 4) thanks to the high photoconductive gain or charge collection efficiency (CCE) up to 15,000% (SI 4), which may originate from the charge injection induced by the surface layer-edge states on 4HPA.

Both the low dark current and highly stable SNR result in a very low level of baseline drift and a highly stable photocurrent output signal. Therefore, the 4HPA detector exhibits superior direct imaging capability (Fig. 6f, g). The high color contrast demonstrates that the object is clearly resolved, furthermore, it shows unambiguously the similarity in shape and size between the nut and the X-ray image which means that there is no variation or deviation. The as-fabricated 4HPA

devices also show good stability after the long-term operation, which is vital for the actual imaging application (Fig. 6h, i and SI 6).

The 4HPA detectors in this work show great potential for implantable or wearable devices used for healthcare monitoring applications, for example, beam monitoring during medical therapy of tumors inside of human bodies, personal dosimetry for people in radiation environments, e.g., radiotherapists and astronauts (Fig. S29).

In conclusion, we developed a biocompatible 4HPA OSCS by low-cost solution method for real-time spectra detection of charged particles with single-particle sensitivity, with an energy FWHM of 36%, $(\mu t)_e$ product of $(4.91 \pm 0.07) \times 10^{-5} \, cm^2 \, V^{-1}$ for the uncollimated 5.49 MeV α particles, and spectra detection time down to 3 ms. The high detection performance may originate from radiation-stimulated "heavy-to-light electron transition" that is influenced by L1 levels and surface-edges-promoted exciton dissociation mechanisms. In 4HPA OSCS with 2D crystallographic structure and discontinuous band structure, the macroscopic charge transport behaviors of radiations excite electrons may be influenced by high-energy L1 level with smaller effective mass than LUMO level, leading to ultrahigh drift velocity up to $5 \times 10^5 \, cm \, s^{-1}$ at "radiation-mode" while ultralow background current (12 pA at an electric field of 100 V $cm^{-1}$) at "dark mode" along the $a$ axis of 4HPA OSCS. In addition, nanoscale surface layer edges on 4HPA OSCS originating from its 2D crystallographic structure induced molecular self-assembly during the solution growth process, also promote exciton dissociation along the in-plane direction ($a$ axis) of this OSCS. Both factors lead to a superior electron transport and radiation detection performance along the $a$ axis of 4HPA OSCS. Thanks to this property, 4HPA detectors can also be used for efficient X-ray detection with a sensitivity of 16,612 μC $Gy_{abs}^{-1} \, cm^{-3}$, detection of limit down to 20 $nGy_{air} \, s^{-1}$, and long-term work stability even after 690 $Gy_{air}$ irradiation.

This work demonstrates that 4HPA detectors the biocompatible and highly stable semiconductor radiation detectors achieving efficient and direct detection of charged particles, X-ray, and fast neutron detection. Together with its compact, highly localized, tissue-equivalent, and low-cost properties, 4HPA detectors are promising for radiation imager for complementary X-ray (sensitive to heavy atoms) and fast neutrons (sensitive to light atoms) imaging and wearable/implantable personal dosimeters for in-vivo and broad-band radiation monitoring during radiotherapy and other medical checking applications.

# Methods

## Materials and crystal growth
4-Hydroxyphenylacetic acid (4HPA, 99%) powder was purchased from *Macklin Reagent Ltd*. Ethylic ether (min. 99.5%) was purchased from *Sinopharm Chemical Reagent Co., Ltd*. 4HPA single crystals are obtained by a controlled solvent evaporation method. The 4HPA powder was purified by dissolving 4HPA in ethylic ether (EE) and then followed by a filtration process. The solvent slowly evaporated to leave some 4HPA crystals on the bottom of the baker, which were then removed from the beaker and washed with warm petroleum ether (PE). 4HPA single crystals are grown from EE solvent with a concentration range from 1 to 8 mg $ml^{-1}$ and a growth temperature of 5–10 °C. Before the solvent entirely flowed away, the transparent and rectangular crystals were taken from the beaker almost without any macro damage. In particular, all steps were carried out in an ultra-clean chamber to prevent any dust pollution.

## Cell culture treatment
All cell experiments were performed in a sterile environment. Materials were purified through a 200 nm filter before the cellular experiment. HUVECs (Human umbilical vein endothelial cells) were cultured in Dulbecco's modified eagle medium (DMEM), including 10% fetal bovine serum (FBS), 60 μg $ml^{-1}$ of penicillin and 100 μg $ml^{-1}$ streptomycin in a humidified incubator (HF90, Heal Force, China) at 37 °C and 5% $CO_2$. The cell viability was evaluated by measuring the mitochondria metabolic activity of HUVECs using a CCK-8 assay. HUVECs were seeded in a 96-well plate in cell medium ($8 \times 10^3$ per well) overnight and subsequently, the supernatant was removed and different doses of 4HPA or 4HCB dispersed in the DMEM were added to the cells and further incubated. After 24 h, a 5% volume of CCK-8 solution in a serum-free medium was added to the cells. After incubation at 37 °C for 1 h, the absorbance of each well (OD 450 nm) was measured by using a microplate reader, and the cytotoxicity of different treatment groups was calculated by the absorbance.

## Structural and thermal behaviors
Powder X-ray diffraction (XRD) measurements were performed by grinding one piece of the as-grown 4HPA single-crystal into fine powder in a mortar. The Bruker D8 Advance equipment with Cu $K\alpha$ ($\lambda = 1.5406$ Å) under a voltage of 40 kV and a current of 40 mA was used; the scanning angle was in the range of 5–40° and the scanning rate was 5° $min^{-1}$. The single-crystal XRD patterns for 4HPA were also measured under the same conditions. The thermal behavior of the as-grown single-crystal was evaluated using a Mettler-Toledo Differential Scanning Calorimeter (DSC), over the range of 20–150 °C at a heating rate of 5 °C·$min^{-1}$ in a nitrogen atmosphere followed by controlled cooling under the same conditions.

## DFT calculation
First-principles calculations were performed by using the projected augmented-wave method[63], which is implemented in the Vienna Ab-initio Simulation Package (VASP)[64]. The generalized gradient approximation (GGA) of the Perdew-Burke-Ernzerhof (PBE) type was employed to treat the exchange-correlation interaction[65]. The kinetic energy cutoff was set to 520.0 eV. During structural relaxation, all atoms were allowed to relax until the Hellmann-Feynman force on each atom was less than 0.01 eV $Å^{-1}$. The relaxation of the atomic position is terminated when the continuous variation of total energy is less than $10^{-5}$ eV. The Brillouin-zone integration was carried out by using $6 \times 4 \times 2$ for the geometry optimization and the total energy at $\Gamma$-centered grids. The effective mass was obtained from the band structure according to the formula $m = \hbar^2 (d^2E/dk^2)^{-1}$ equation[66].

## Monte Carlo simulation
The simulation is conducted by Geant4 package using $C^{++}$ toolkit, which can probe the interaction between materials and incident particles[67]. For the Geant4 simulation, the geometry (G4VUserDetectorConstruction) is defined as 4HPA, $C_8H_8O_3$, density of 1.253 g $cm^{-3}$, size of $6 \times 6 \times 10 \, cm^3$; the particle beam source (G4VPrimaryGeneratorAction) is defined as linear α beam with diameter of 2 mm, energy of 5.49 MeV, penetrating direction perpendicular to the 4HPA surface; the physical interaction (G4VUserPhysicsLists) is FTFP_BERT. The results are the initial energy, penetrating depth, and energy deposit for each particle-material interaction.

## Optical property characterization
The UV–Vis transmission spectra of 4HPA single-crystal were measured on a UV-3150 spectrometer (Shimadzu, Japan) in the range 800–200 nm at room temperature. These spectra were used to generate the absorption and thus determine the bandgap using a linear fit of the Tauc plot.

## Electrical property characterization
Both the sandwich and coplanar electrode configurations were fabricated with gold (Au) electrodes by a thermal evaporation method. Au

electrode thickness is around 100 nm. *I–V* measurements at room temperature were performed using a Keithley 6517 Picoammeter and a stabilized bias supply. For the X-ray photocurrent, an X-ray tube was used, with an accelerating voltage of up to 50 kV and tube current increasing from 5 to 50 μA. Photocurrent data was acquired using the Keithley 6517 Picoammeter with current–time (*I–T*) mode when the bias voltage was constant.

## Alpha particle mobility and spectra detection

The 4HPA detector was positioned inside a copper chamber at a distance of 1 cm from the $^{241}$Am alpha particle source. During the measurement process, we shielded the whole device using a paper with thickness of 2 mm, except for a hole (diameter of 3 mm) to irradiate the alpha particles to the cathode or anode of 4HPA detectors to avoid any possible ionization from the other objects or wire.

The detector was powered by a high-voltage power supply (ORTEC, 556), then the output signal was sent to the pre-amplifier (ORTEC, 142PC) and amplifier (ORTEC, 572 A) for signal amplification. For certain detectors, the electrons-only signal and hole-only signal can be measured by applying positive and negative bias voltages. At each applied electric field, an average signal pulse (hole-only or electrons-only) that reflects the carrier drift time and pulse height, was obtained by averaging 200 pre-amplifier single pulses induced by alpha particles. The average pulse rise time was measured to calculate the electrons or hole drift time. The amplified signal was input to a digital analysis for data processing with an alpha particle spectrum output.

## X-ray detection and imaging

For X-ray detection, we used an Amptek Mini-X tube to generate an X-ray beam. During the experiment, a 0.15-mm thick Cu attenuator was used to weaken the incident X-ray dose rate, the distance between the X-ray source and 4HPA detectors was 10 cm. The X-ray source was operated with a tube voltage of 40 kVp (X-ray photo energy up to 40 keV and peak intensity at 27 keV) and adjustable tube current in the range of 0.005–0.06 mA to control the dose rate of 0.35–2.34 μGy$_{air}$ s$^{-1}$. For X-ray irradiation, the Spellman X4060 X-ray generator was used. The largest dose rate of 19.18 mGy$_{air}$ s$^{-1}$ was obtained with a tube voltage of 150 kVp and a tube current of 1.2 mA. The total irradiation time is a continuous 10 h for each 4HPA detector. The X-ray dose rate was carefully calibrated by using a Fluke Si diode (RaySafe X2) dosimeter. The Keithley 6517B was utilized here for applying the high bias voltage and measuring the dark and photocurrent with high precision. All the measurements were conducted in air condition and at room temperature.

## Data availability

The data that support the findings of this study are present in the article and Supplementary Information. Additional data related to this study are available from the corresponding author upon request.

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

## Acknowledgements

This work is supported by the National Key Research and Development Program of China (2023YFE0206300) and the National Natural Science Foundations of China (No. U2032170, 11922507, and 11975121). This

project is also supported by the Natural Science Foundation of Shaanxi Province (2020JC–12), the Natural Science Basic Research Plan in Shaanxi Province of China (2021GXLH-01-03), and the Fundamental Research Funds for the Central Universities (3102020QD0408). The authors thank Prof. Yang Yang from Zhejiang University for helpful discussions.

## Author contributions

D.Z. grew 4HPA single crystals, performed the electric measurements, and wrote the manuscript with inputs from all authors. R.G., C.Z., and S.Y contributed to DFT calculations and data analysis. W.C. contributed to Monte Carlo simulations. M.W. and L.S. performed the cell incubation experiments. X.Z. assisted in structural characterizations. T.Y. and P.S. conducted data analysis and organic semiconductor theory. T.S. and W.J. advised on experiments. Y.X. supervised the project.

## Competing interests

The authors declare no competing interests.
