## [Peer Review File · Nature Communications]

REVIEWER COMMENTS

Reviewer #1 (Remarks to the Author):

The authors studied the 4HPA OSCS and claimed excellent properties. However, characterization with radiation for the radiation monitoring is not properly studied. Authors claim of “heavy-to-light electron transition” based on VASP theoretical study need to be proved by the experiment since it is not commonly acceptable concept in radiation detection in crystals. Also some of characterization method and interpretation of results are not well described which may lead a misleading. Since it lacks of the above study, I don't recommend this manuscript for publication in nature communication.

The following a major comment

1. Authors claim of “heavy-to-light electron transition” based on VASP theoretical study is based on statistical model which is not happening in neither semiconductor or scintillation crystals since hot electron and holes are produced in the high energy radiation will be cascade to lower energy electrons and holes, and it become thermalized by phonon scattering in pico-second level unless there is a forbidden mechanism. Thus it should be happen the same way in the OSCS. Why L1 level electron can not be thermalized by phonon coupling and move to HOMO level before reach to the electrode? For this claim, authors need to inject monochromatic pulse of photon energy from 3.8 to 6 eV and check whether or not there is dramatic change in L1 level.

2. For the radiation detection, the following study is required more. a) X-ray detection efficiency. b) number of electron per keV radiation c) radiation hardness.

a) One of the problem with OSCS is low detection efficiency of X-ray since low Z and density. Of course, there could be a merit of OSCS if effective Z and density is compatible with tissue but 4HCB does not looks like the case.

b) Why alpha energy resolution is so bad compare with CZT or Si?

c) OSCS looks like there is a degradation with radiation. For the clinical application it is crucial. How about 4HPA case?

3. It is not enough to only show alpha response since 4HPA is not aiming for the alpha detector (Si is much better performance than 4HPA). For the comparison with 4HCB or others, it is absolute necessary to show minimum detection dose rate such as uGy-air/sec or uC Gy-air/sec.

4. I don't see authors claim of advantage of biocompatible since it should have a electrode, wire for the HV and preamp to avoid noise should be included for the clinical application even if 4HPA is a biocompatible.

- 5) For claimed application, low voltage is required. For example Si sensor operate lower voltage and it is robust, much better performance.
- 6) For the clinical application ion-chamber and Si sensors are being used commonly authors need to compare the performance of 4HPA with that.
- 7) Authors claim wrote “normally contains a fast-rising (below 1 μ s) and an accompanied slow-rising components (tens of microseconds) in organic detectors while that usually below 1 μ s in inorganic detectors, e.g., CdZnTe 15,17 . Whether seeking novel materials or possible solutions to improve the charge transport in OSCS” It is not inorganic semiconductor property but people are using the semiconductor which has electron and hole mobility are similar such as CZT, CT, Si, HPGe. If either electron or hole mobility is very low like 4HCB or 4HPA, it is difficult to use for spectroscopic application since there is slow rising time which depends on the position. Thus authors need to explain how 4HPA can get good performance when hole mobility is much lower than electron mobility.
- 8) Authors need to show fitting results of alpha particle energy resolution. It is not Gaussian and it shows a shoulder and tail, why? See Ref 16 for the comparison.
- 9) Authors should not use not well-defined quantity such as “still requires a long duration time around 30 minutes for a spectrum measurement” or 0.5-10 sec in Fig 5 since does rate is not known
- 10) Authors claimed as “ 4HPA detectors exhibit the record spectra detection of α particles among their organic counterparts, with energy resolution of 36%, (μ t) e product of $(4.91\pm 0.07)\times 10^{-5}$ cm² V⁻¹ , and spectra detection time down to 3 ms” As I mentioned spectral detection time down to 3ms is not meaningful unless the total alpha rate is described. (with a few alphas, Si can detect it with good energy resolution) According to Ref 21, (μ t)h = 8.5×10^{-5} cm²/V which is better than authors results. Even if they don't give an energy resolution of an alpha particle, it looks better than authors results at HV.
- 11) Geant4 is not the best tool for alpha particle simulation, SRIM is recommended in this application.
- 12) In the clinical application with X-ray or gamma-rays, it can not be used for the spectroscopic mode since intensity is high, thus authors need to also show integration mode with X-ray.

Reviewer #2 (Remarks to the Author):

The spectra detection of charged particles exhibiting single-particle sensitivity is common in the realm of particle detection. Achieving 100% efficiency in detecting charged particles is always possible; the pivotal factor lies in the effective collection of the particle's charge, i.e., CCE, which directly impacts energy resolution. Silicon (Si) detectors routinely attain an energy resolution as fine as 1%. The reported energy

resolution of 36% by this work is unimpressive. This remarkable of this work is attributed to the device's flexible format for wearable devices, which holds a distinctiveness worthy of publication. BTW, the usage of the term "energy FWHM 36%" is inaccurate. Equally remarkable is the measured resistivity of $(1.28 \pm 0.003) \times 10^{12} \Omega \cdot \text{cm}$ from this work.

"For the first time, the charge transport mechanism responsible for the superior performance is clarified in view of high-energy physics".

I was not quite understood this statement. Firstly I would never claim "for the first time", we don't know what we don't know. What has the high energy physics to do with this device? What energy range is considered the domain for high energy physics? To my knowledge, it is the order of magnitude of the energy dealt with by this paper.

"Geant4 simulation indicates that 5.49 MeV α particles 154 can ionize electrons with energy of 0-150 eV (Fig. S9),"

How is zero energy possible for ionized energy?

Fig 3c should be just a Bragg peak, but it doesn't seem like the SRIM/TRIM simulation of the stopping power curve. I would double-check and verify Geant 4 simulation.

"The 4HPA detector was positioned inside a copper chamber at a distance of 1 cm from the 241 Am alpha particle source. "

Is the copper chamber under vacuum? Are there any wire near the radiation field, in other words, how do you rule out the signal is not coming from air ionization?

Reviewer #3 (Remarks to the Author):

This paper reports the solution-grown biocompatible n-type organic single-crystalline semiconductor materials for real-time spectra detection of charged particles with single-particle sensitivity, finding a “heavy-to-light electron transition” dominated charge-transport mechanism for organic radiation detectors. This novel radiation detector possesses excellent biocompatibility (cell viability over 90% after 24-hour incubation concentration of 2 mg ml⁻¹), which is comparable to the state-of-the-art carbon-based biocompatible materials. In the meanwhile, the detector achieves recorded charge-particle detection performance within their organic counterparts, with energy resolution of 36%, spectra detection time down to 3 ms, and mobility-lifetime product of electrons of $(4.91 \pm 0.07) \times 10^{-5} \text{ cm}^2 \text{ V}^{-1}$. Furthermore, the author combines DFT, Monte Carlo simulation, and time-dependent electrical measurements, proposing a new charge transport mechanism for organic radiation detectors, which is very interesting and significant to understand the charge-transport behaviors of organic semiconductors under irradiation and design new organic radiation detectors.

Overall, this paper demonstrates the first OSCS detector as low-cost consumer electronics for wearable/implantable dosimeters with real-time, position-sensitive, and in-vivo healthcare monitoring in radiation-exposure environments; and broadens the charge-transport theory of organic semiconductor from the viewpoint of high-energy physics. Therefore, I recommend this paper to publish on Nature Communications after addressing following comments.

1. What are conventional detector materials for in-vivo radiation monitoring? Compared with the conventional material, the benefits to use 4HPA organic detectors should be described.
2. The 4HPA detectors show better detection performance and biocompatibility than the reported 4HCB detectors. What is the difference of 4HPA and 4HCB molecules? How does it influence the physical properties of these two materials?
3. In this paper, the author reports a very simple solution method for 4HPA single crystal growth. Since the crystal quality is very important for detection performance, how does the author control the quality of 4HPA single crystals?
4. What is the effective thickness of 4HPA semiconductors for charged particles detection? Is it possible to fabricate flexible 4HPA detectors for wearable/implanted devices?
5. How about the work stability of 4HPA detectors?
6. In Fig. S15, the author also gives the result of direct fast neutron detection using 4HPA detectors. Why does the 4HPA OSCS can detect X-rays and fast neutrons? More comments should be given on the potential applications of this properties.

Reviewer #1 (Remarks to the Author):

The authors studied the 4HPA OSCS and claimed excellent properties. However, characterization with radiation for the radiation monitoring is not properly studied. Authors claim of “heavy-to-light electron transition” based on VASP theoretical study need to be proved by the experiment since it is not commonly acceptable concept in radiation detection in crystals. Also some of characterization method and interpretation of results are not well described which may lead a misleading. Since it lacks of the above study, I don't recommend this manuscript for publication in nature communication.

Answer: We thank for the reviewer's insightful comments and feedback. We have revised the manuscript according to the comments.

The following a major comment

1. Authors claim of “heavy-to-light electron transition” based on VASP theoretical study is based on statistical model which is not happening in neither semiconductor or scintillation crystals since hot electron and holes are produced in the high energy radiation will be cascade to lower energy electrons and holes, and it become thermalized by phonon scattering in pico-second level unless there is a forbidden mechanism. Thus it should be happen the same way in the OSCS. Why L1 level electron can not be thermalized by phonon coupling and move to HOMO level before reach to the electrode? For this claim, authors need to inject monochromatic pulse of photon energy from 3.8 to 6 eV and check whether or not there is dramatic change in L1 level.

Answer: We appreciate for the reviewer's comment. In inorganic semiconductors, radiation excited hot electrons and holes become thermalized by phonon scattering in around pico-second level. It should be happening the same way in OSCS. **We didn't claim the alpha-particle ionized electrons could not be thermalized by phono scattering.**

In this paper, we used “**heavy-to-light electron transition**” model to explain the difference between our theoretical and experimental results. This model could be described as **the radiation excited electrons may have a certain proportion on L1 level or**

L1 level may affect the macroscopic charge transport when the two conditions are satisfied at the same time: High-energy radiation are continuously injected into 4HPA detectors and high electric field is continuously applied to 4HPA detectors.

In our experimental work, we conducted the electron-only and hole-only charge transport **experiments** by alpha particle time-of-flight (α -TOF) method. We found **whatever along a or c axes, electrons generally have better charge collection efficiency and faster drift time than holes at the same measurement condition and same sample (Fig. R1),** that means the electrons mobility-lifetime products and mobility should larger than that of holes. However, according to the **DFT calculation (Fig. R2 and Table R1), HOMO level has smaller effective mass (m^*) than the LUMO level in 4HPA along both c and a axis which is contradictory with our experimental result (Table R2), while the **H1 level has larger m^* than L1 level,** which is consistent well with the experimental result.**

Fig. R1 Charge carrier transport behaviour in 4HPA single crystal along the c axis. (a) Electron-only charge rising pulses (averaged by 100 single electron-only pulses) under a series of electric fields, (b) hole-only charge drift time under a series of electric fields.

Fig. R2 Band structure of 4HPA organic single crystal semiconductor. (a) The energy band structure, (b) anisotropic $1/m^*$ along a and c axes of 4HPA OSCS.

Table. R1 Anisotropic $1/m^*$ value along the a and c axes of 4HPA

$1/m^*$	HOMO	H1 Level	LUMO	L1 Level
a axis	3.06	3.79	1.38	4.98
c axis	2.14	0	0.92	1.47

Table. R2 Anisotropic charge mobility in 4HPA single crystal ($\text{cm}^2 \cdot \text{V}^{-1} \cdot \text{s}^{-1}$)

Mobility	Electron	Hole
a axis	4.17 ± 0.04	N/A
c axis	2.60 ± 0.05	0.06 ± 0.001

In addition, we observed the remarkable “fast-rising” and “slow-rising” components in each signal pulse of 4HPA organic detectors under irradiation of high-energy alpha particle (Fig. R3), which was hardly explained by conventional de-trap mechanism. If we think the “slow-rising” stage is due to the de-trap process, that means a proportion of electrons on LUMO level can transport across the 4HPA detectors (0.25 cm) in the “fast-rising” period, which is down to 50 ns when the electrode field of 2800 V cm^{-1} (Fig. R4(a)). Then we can roughly estimate the charge transport mobility as $1790 \text{ cm}^2 \text{ V}^{-1} \text{ s}^{-1}$ at irradiation and high bias voltage conditions, which sounds impossible in 4HPA case because its high resistivity (Fig. R4(b) $\sim 10^{12} \Omega \text{ cm}$, rely on carriers from HOMO and LUMO levels) under dark condition.

Fig. R3 Typical (from down to up) time-resolved average electron-only drift pulses from α -TOF method measurement along a axis of 4HPA with bias voltages from 800 V cm^{-1} to 4000 V cm^{-1} .

Fig. R4 Charge transport properties of 4HPA detectors (a) the fast-rising component in single electron-only pulse under alpha-particle irradiation with bias voltage of 2800 V cm^{-1} along the a axis, (b) current density vs. applied electric field curve under dark condition.

In further, if we compare the time-resolved average electron-only/hole-only drift pulses from α -TOF method along the c axis (**Fig. R1**, c axis is less affected by the surface layer edge states), we can found that (at the same measurement condition) **electrons transport time is longer than that of holes at the low bias voltage and becomes much faster at the high bias voltage**, which may indicate the L1 level may affect the macroscopic charge transport with the high bias voltage condition.

Finally, we consider the energy distribution of alpha-particles ionized electrons, Fig. R5. Even after multiple ionization, the electrons also have high energy up to 150 eV (Fig. R5). It is possible for electron to occupy the L1 level (6 eV) before thermal relaxation process. Although thermal relaxation time from high-energy level to low-energy level should be very fast in inorganic semiconductors, here the band gap between L1 level and LUMO level (Fig. R6), continue irradiation and high bias voltage may prolong the thermal relaxation time, which may lead to a small proportion of electrons on L1 level or L1 level may still has influence on macroscopic charge transport properties in 4HPA detectors.

Fig. R5 Energy distribution spectra of electrons ionized by 5.49 MeV α particles. (a) alpha particles ionized electrons, (b) second ionized electrons, (c) multiple ionized electrons.

Fig. R6 Band gap between L1 and LUMO levels.

To prove this model, we considered the experimental/calculated method: Measure/Calculate the electrons' thermal relaxation time from L1 level to LUMO level in 4HPA detectors under continuous high-energy irradiation and high electric field conditions. However, both methods are very difficult based on the current

experimental/calculation facilities. We will try to find other possible equipment or other method to prove this mechanism.

As a conclusion, we think **L1 level influence at continuous irradiation and high electric field may be one possible explanation to our result**. According to current organic solid theory, we cannot totally prove or deny this influence from L1 level. But this will be an interesting and open topic for us to exploration in the future. **After careful consideration and discussion with our all co-authors, we decided to revise the related content in the manuscript to avoid any possible misunderstanding.**

Revised main manuscript:

We revised abstract “**Macroscopic transport behavior of radiation-excited electrons in 4HPA may influenced by high-energy L1 level with smaller effective mass than LUMO level**” in Page 1 and 2.

We deleted “**This suggests an extraordinary benefit inorganic radiation detectors.**” and **Fig. 2(e) and (f)** in main manuscript, Page 6, which may lead misunderstanding.

We revised main manuscript “**In addition, Fig. 3(d) and (e) give the Monte Carlo simulated energy spectra of α particles excited electrons after first ionization and multiple ionization, respectively, indicating that radiation generated electrons possess energy of 0.8~150 eV before the thermal relaxation.**” in Page 6.

We revised main manuscript “**This suggests that macroscopic charge transport behaviors maybe affected by L1 level, or the ionized electrons may also have a certain proportion on L1 level even after the thermal relaxation process, when two conditions that the high-energy radiations are continuously injected into 4HPA detectors and high electric field is applied are satisfied at the same time. Then fast-rising and slow-rising stages in the rising pulse may originate from hybrid influence from L1 and LUMO levels, respectively.**” in Page 7.

We revised main manuscript “ **α particles produced electrons maybe affected by L1 level, leading to “heavy-to-light electron transition” macroscopic transport behaviors**” in Page 10.

We revised main manuscript “the macroscopic charge transport behaviors of radiations excite electrons maybe influenced by high-energy L1 level with smaller effective mass than LUMO level” in Page 14.

2. For the radiation detection, the following study is required more. a) X-ray detection efficiency. b) number of electron per keV radiation c) radiation hardness.

a) One of the problem with OSCS is low detection efficiency of X-ray since low Z and density. Of course, there could be a merit of OSCS if effective Z and density is compatible with tissue but 4HCB does not looks like the case.

b) Why alpha energy resolution is so bad compare with CZT or Si?

c) OSCS looks like there is a degradation with radiation. For the clinical application it is crucial. How about 4HPA case?

Answer: We appreciate for the reviewer’s comment. Theoretical and experimental X-ray detection efficiency, number of electrons per keV radiation, and radiation hardness are added.

a). X-ray detection efficiency of 4HPA detector is determined by the **X-ray attenuation efficiency (ϵ)** and **Charge Collection Efficiency (CCE)**.

4HPA detectors have smaller X-ray attenuation efficiency (**Fig. R7**, 2 mm-thick 4HPA detectors have ~100 % for 10 keV X-ray, ~5.6 % for 40 keV X-rays) when compared with the inorganic detectors with high atomic numbers (**Z**), but the **CCE** of 4HPA X-ray detectors is up to **15000%** (**Fig. R10(b) and (d)**) due to the high photoconductive gain that may originate from the surface layer edge states. It indicates that the overall X-ray detection performance of 4HPA is comparable with inorganic X-ray detectors.

We characterized the X-ray detection performance of 4HPA detectors (size: 0.003 cm³) for 40 kVp X-ray source. X-ray Detection **Sensitivity (S)** is up to 304 $\mu\text{C Gy}_{\text{air}}^{-1} \text{cm}^2$ or 16612 $\mu\text{C Gy}_{\text{abs}}^{-1} \text{cm}^3$ (**Fig. R10(c) and (e)**), and the **Detection of Limit (LoD)** is down to 20 $\text{nGy}_{\text{air}} \text{s}^{-1}$ (**Fig. R16**).

It worth note that the 4HPA detectors possess very low dark current of 0.1 pA at applied bias voltage of 150 V, and very low dark current drift of $1.09 \times 10^{-8} \text{ nA cm}^{-1} \text{ s}^{-1} \text{ V}^{-1}$ (**Fig.**

R11), which is very significant to increase the **Signal-to-Noise Ratio (SNR)**, especially for organic semiconductor detectors with weak X-ray absorption. We also calculated the **Sensitivity/Dark current density (S/J_{Dark})** value of 4HPA detectors, which is up to $10^4 \mu\text{Gy}_{\text{air}}^{-1} \text{s}$ and is much better than 4HCB and some inorganic X-ray detectors like a-Se, MAPbBr₃, Cs₂AgBiBr₆, etc (**Fig. R16**).

We added the following part in **SI file**:

a) 1.1. X-ray attenuation efficiency (ϵ)

We added following part in **SI 4**.

“SI 4 X-ray Attenuation Efficiency (ϵ) of 4HPA Detector

For the X-ray beam with single photon energy, the attenuation efficiency¹,

$$\epsilon(E) = 1 - \frac{I}{I_0} = 1 - e^{-\mu(E) \times \rho \times x} \quad (\text{S3})$$

where I is the transmitted intensity, I_0 the incident intensity, $\mu(E)$ is the total attenuation mass coefficient from XCOM: Photon Cross Sections Database² (**Fig. S22(a)**), ρ is the mass density of 4HCB detector and x is the thickness of the detector. As shown in **Fig. S22(b)**, 2 mm-thick 4HPA detectors possess 100% absorption rate for X-rays below 10 keV, and the same attenuation efficiency of human tissue that indicates the very good tissue equivalence compared with other detectors with high atomic numbers like Si and CdTe.”

Fig. R7 (Fig. S22) X-ray absorption of 4HPA single crystal. (a) Total attenuation cross section comparison 4HPA, 4HCB, and Si, (b) X-ray attenuation efficiency of 4HPA detectors with series of thickness changes with incident photon energy.

a) 1.2. Charge collection efficiency & number of carriers per eV radiation

We added following content in **SI 5**.

“SI 5 Charge Collection Efficiency and Detection Sensitivity of 4HPA Detectors

For 4HPA organic X-ray detectors, the **Charge Collection Efficiency (CCE)** was calculated by¹:

$$CCE = \frac{I_R}{I_P} \quad (S4)$$

where I_R and I_P are measured and theoretical photocurrent under X-ray irradiation. I_P is defined as $I_P = \varphi\beta e$, where φ is the photo absorption rate, β is the maximum number of photo generated carriers per photon, and e is the elementary charge. The photo absorption rate is given by $\varphi = \varepsilon D m_s / E_{ph}$ photons s^{-1} , where ε is the attenuation efficiency, D is the dose rate, m_s is the sample mass, E_{ph} is the energy per photon¹.

➤ number of carriers per eV radiation

In particular, β , the maximum number of carriers generated by per photon, which is estimated as $\beta = 2 \times \frac{E_{ph}}{\Delta E}$, ΔE is the electron-hole creation energy, is estimated as 3 times band gap ($E_g=3.8$ eV) of 4HPA³, (around 175 electrons and holes generated by 1 keV photon).

Then, we calculated and performed theoretical and experimental X-ray photocurrent induced by 4HPA detectors, respectively. The tube voltage of X-ray is 40 kVp and the energy of X-ray photons in range of 0 ~ 40 keV. The estimated energy distribution of X-ray beam by using the software (SPEKTR 3.0), as shown in **Fig. S23**. SPEKTR 3.0 is kindly shared by the **I-STAR** lab at John-Hopkins University and can be freely downloaded from <http://istar.jhu.edu/downloads/>.

For theoretical photocurrent measurement, we need to calculate the equivalent absorption efficiency ($\bar{\varepsilon}$) because the 4HPA detector has different attenuation efficiency ($\varepsilon(E)$) between photons with different energy. First, the distribution of photons energy ($N(E)$, red area) and the corresponded absorption rate ($\varepsilon(E)$, blue dots) are shown in **Fig. S24(a)**, where the red filled area is the total energy of incident 40 kVp X-ray beam. Then, the total absorbed X-ray by the 2 mm-thick 4HPA detector is equal to

$$\int_{10}^{40} E \times \varepsilon(E) dE \quad (S5)$$

as shown in **Fig. S24(b)** (red area). Finally, the equivalent fraction of absorbed photons ($\bar{\varepsilon}$) for 40 kVp X-ray beam in 4HPA detectors is calculated⁴,

$$\bar{\epsilon} = \frac{\text{Total absorbed energy}}{\text{Total incident X-ray energy}} \quad (\text{S6})$$

which is around 10% for a 4HPA single crystals of 2 mm thickness under 40 kVp X-ray irradiation.

Then, we measured the Photocurrent-Time (I_R -t) curves of 4HPA detector (Size: 0.003 cm^3) with X-ray switching “ON/OFF” and decreased dose rate of $0.35 \sim 2.34 \mu\text{Gy}_{\text{air}} \text{ s}^{-1}$ under a bias voltage of 5~130 V (**Fig. S25(a)**). The I_P values corresponding to the measurement dose rate were also calculated, with compared with I_R (**Fig. S25(b)**). Then, the CCE values were calculated by I_R/I_P , which is up to 15000% at 150 V of bias voltage (**Fig. S25(c)**). The CCE larger than 100% normally due to the charge injection from electrodes, which may originate from the layer edge states on 4HPA surface. Due to the high CCE values, the 4HPA detectors can achieve high Sensitivity (S) of $330 \mu\text{C Gy}_{\text{air}} \text{ cm}^{-2}$ or $16612 \mu\text{C Gy}_{\text{abs}}^{-1} \text{ cm}^{-3}$ (**Fig. S25(d-e)**) for 40 kVp X-ray detections. Then, the X-ray detection Sensitivity (S)/Dark current density (J_{Dark}) was calculated, which was as high as $1.5 \times 10^{14} \mu\text{Gy}_{\text{air}}^{-1} \text{ s}$ (**Fig. S25(f)**).⁵”

Fig. R8 (Fig. S23) Incident X-ray energy spectrum simulated by SPEKTR.

Fig. R9 (Fig. S24) Effective attenuation efficiency (ϵ) of 4HPA detectors for 40 kVp X-rays. Incident X-ray energy spectrum simulated by SPEKTR (a) Energy distribution (red area) and fraction of absorption (Blue dots) of 40 kVp X-ray photons, (b) total absorbed X-ray energy by 4HPA single crystal under 40 kVp X-ray irradiation.

Fig. R10 (Fig. S25) X-ray detection response of 4HPA single crystal detectors. (a) I-t curve with reduced incident X-ray dose rate, (b) I_R and I_P changes with incident X-ray dose rate, (c) calculated sensitivity of 4HPA detectors. The effective detector size is 0.003 cm^3 , thickness is 0.19 cm , bias voltage is 50 V . The Tube current of X-rays is 40 kV , tube current changes from $5\sim 60 \mu\text{A}$, 0.15 mm Cu is used as attenuator.

We also added following description in **Fig. S19**.

“4HPA detectors also show very low dark current below 0.1 pA at the bias voltage of 150 V , resulting in extremely low dark current drift $\sim 10^{-9} \text{ nA cm}^{-1} \text{ s}^{-1} \text{ V}^{-1}$ (**Fig. S19**). The low dark current drift and high S/J_{Dark} value indicate the highly stable X-ray response and high signal-to-noise ratio of 4HPA OSCS.”

Fig. R11 (Fig. S19) Dark current fluctuation in 4HPA detectors with bias voltage of 150 V.

d). Why alpha energy resolution is so bad compare with CZT or Si?

There are many factors influence the energy resolution of 4HPA detectors for alpha particle spectra when compared with CZT or Si.

- The most important one is 4HPA detector materials should be in further optimized. For radiation detection application, CZT single crystal with ultrahigh purity (~99.99999%) raw materials and advanced melt method for single crystal growth are utilized to avoid undesired carrier traps and defects that have significant for charge transport properties and energy resolution of alpha particles^{6,7}. Si also has same condition.

However, the research period of 4HPA OSCS is much shorter than CZT and Si, there are plenty of space to improve the purity (currently we use 99% raw materials), crystallization quality, defects, and other crystal processing techniques.

- Second reason is due to the limits of charge carrier mobility-lifetime products of organic materials comparing with the state-of-the-art inorganic CZT and Si semiconductors. But we don't want to replace the CZT or Si detectors using the organic detectors, **we would like to dig out the new opportunities of healthcare monitoring applications by organic detectors, for example, lightweight, biocompatible, flexible, tissue equivalent and low-cost personal dosimeters with satisfactory detection performance.** Although the energy-resolution of 4HPA is not the best comparing with the state-of-the-art inorganic Si detectors, but better organic detectors can be further developed by inexhaustibleness molecular engineering methods, the better materials with higher energy resolution could be

expected in the future.

- Third reason is we also need optimize the device electrode configuration and measurement electronics. The electrodes structures and measurement system also have very significant influence on the energy resolution of CZT and Si detectors. For example, Si has low resistivity that can't be used for alpha particles detection, only the Si with diode device structure and reduced leakage current can be used for alpha particle detection. Electrode geometric design also is important for the energy resolution of CZT. At the same time, the measurement electronic systems are also optimized for CZT and Si. **These techniques have been developed tens of years to achieve the high resolution of CZT and Si.**

At present, we have very short research timeslot to develop the ideal electrode structures and measurement electronics. In this work, we only use simple metal electrodes, simple planar electrode configuration, and the measurement system mainly designed for CZT and Si detectors. Therefore, the charge collection efficiency for alpha-particle spectra and signal-to-noise ratio due to the electronic noise is not ideal for 4HPA, leading to low energy resolution (**but the 4HPA result also is the best one among organic detectors**).

In conclusion, **the aim of this work is demonstration of a world-first material with satisfactory overall performances and new opportunities for radiation monitoring sensors**, such as efficient detection performance, broad-band radiation detection (X-ray, charged particles, and fast neutrons), high stability, biocompatibility, light weight, flexibility, and low-cost solution methods for wearable and implantable sensors. In the future, there is a plenty of space to improve the energy resolution of such organic detectors by various molecular engineering, optimization of the materials purity, crystallization quality, device structure and measurement electronics.

We revised main manuscript “.....measured spectra (electrons-only signal) by coplanar 4HPA detectors ($E//a$ axis) at a series of bias voltages, with the best energy resolution (**ER**) around 36% among organic detectors. Although the **ER** of 4HPA detectors still not very good when compared with the state-of-the-art inorganic detectors, e.g., CdZnTe and Si, there is a plenty of space to improve the energy resolution of such organic detectors by various molecular engineering, optimization of the materials purity, crystallization quality, device structure and measurement electronics.” in Page 11.

c) OSCS looks like there is a degradation with radiation. For the clinical application it is crucial. How about 4HPA case?

4HPA OSCS have very good work stability for radiation detection. The photodetection performances didn't show any degradation after irradiated by 150 kVp, 690.48 Gy_{air} X-ray.

We added following content in **SI 6**.

“SI 6 Degradation with radiations

At first, we evaluated the long-time **I-t** curves with “ON/OFF” switching behaviors of 4HPA detectors. With the increasing X-ray dose rate, as shown in **Fig. S26(a)**, after 50 cycles (continuous operation of 3100 s), the 4HPA detectors didn't show any degradation. Then, we measured the 100 “ON/OFF” switching cycles (continuous operation of 6000 s) under constant X-ray irradiation with dose rate of 19.18 mGy_{air} s⁻¹ and bias voltage of 150 V. No degradation happens in both dark and photocurrent (**Fig. S26(b)**).

In addition, long-time current drift stability measurements of 4HPA devices were also carried out with continuous 2-hour dark current measurement and photocurrent measurement with X-ray irradiation (dose rate of 19.18 mGy_{air} s⁻¹, total dose of 138.096 Gy_{air}). The result shows both the dark current drift (around 10⁻⁹ nA cm⁻¹ s⁻¹ V⁻¹) and photocurrent drift are very small, and no degradation occurs after continuous 2-hour device work period (**Fig. S27**).

In further, we utilized 150 kVp X-ray (dose rate of 19.18 mGy_{air} s⁻¹) to irradiate 4HPA detectors for 10 hours, with total irradiation dose of 690.48 Gy_{air} (upper limit of our X-ray generator). We compared the photodetection properties of before and after 10-hour irradiation, as shown in **Fig. S28**. Both dark current and photocurrent didn't show any obvious degradation.

These results indicate 4HPA detectors show very good radiation stability and long-term work stability at high electric field.”

Fig. R12 (Fig. S26) X-ray detection long-term work stability (a) X-ray photocurrent I-T cycles change with increased dose rate, (b) X-ray photocurrent I-T cycles when the work time continues to 6000 s with bias voltage of 150 V.

Fig. R13 (Fig. S27) X-ray detection long-term work stability of 4HPA detectors with continue 120 min work for X-ray detection.

Fig. R14 (Fig. S28) X-ray detection long-term work stability after continuously detection of 0 h, 9 h, 1 h, and 10 h and 150 kVp X-ray dose rate of $19180 \mu\text{Gy}_{\text{air}} \text{s}^{-1}$, total irradiation dose of $690.48 \text{ Gy}_{\text{air}}$.

- It is not enough to only show alpha response since 4HPA is not aiming for the alpha detector (Si is much better performance than 4HPA). For the comparison with 4HCB or others, it is absolute necessary to show minimum detection dose rate such as $\mu\text{Gy}_{\text{air}}/\text{sec}$ or $\mu\text{C Gy}_{\text{air}}/\text{sec}$.

Answer: We appreciate for the reviewer's comment. The following description related to the minimum detection dose rate—**Detection of Limit (LoD)** of 4HPA detectors was added in **main manuscript** and **SI 3**.

Main manuscript:

“It is noteworthy that the stable and repeatable detectable dose rate $0.35 \mu\text{Gy}_{\text{air}} \text{s}^{-1}$ has been experimentally achieved (**Fig. S20**). The **Signal-to-Noise (SNR)** ratio of photocurrent under each dose rate with each bias voltage were calculated (**Fig. S21**), by which the **Detection of Limit (LoD)** values at each bias voltage (**Fig. 5(b)**) were estimated according to the SNR value larger than IUPAC⁸ defined 3. The best **LoD** of $20 \text{ nGy}_{\text{air}} \text{s}^{-1}$ (275 times lower than required $5.5 \mu\text{Gy}_{\text{air}} \text{s}^{-1}$ for regular medical diagnostics⁹.”

“SI 3 Detection of Limit of 4HPA detectors

Fig. S20 shows the typical **I-t** curves and corresponding calculated **Signal-to-Noise Ratio (SNR)** values of the 4HPA detectors (**Fig. S21(a)**). Although $0.35 \mu\text{Gy}_{\text{air}} \text{s}^{-1}$ is the lowest dose rate in our measurement system, we estimated the minimum dose rate according to the **SNR** value larger than 3, which is around $0.02 \mu\text{Gy}_{\text{air}} \text{s}^{-1}$ ($20 \text{ nGy}_{\text{air}} \text{s}^{-1}$), as shown in **Fig. S21(b)**. The **LoD** for 4HPA is lower than reported value for 4HCB

with $0.29 \mu\text{Gy}_{\text{air}} \text{ s}^{-1}$. We compared the **LoD** and **S/J_{Dark}** of 4HPA, 4HCB and other halide perovskite single crystalline detectors (**Fig. S21(c)**)^{5,10-17}. Although organic semiconductors normally have weaker X-ray absorption efficiency and lower charge mobility than inorganic single crystalline semiconductors, we achieved the excellent X-ray detection performances comparable with halide perovskites that possess benefits of high-Z, high charge mobility, and solution-processing method. In addition, compared with halide perovskite, 4HPA organic detectors also have better tissue-equivalent, biocompatible, and flexible properties, which make them promise for next-generation lightweight and human-friendly wearable or implantable X-ray dosimeters.”

Fig. R15 (Fig. S20) X-ray detection photocurrent of 4HPA single crystal detectors changes with incident dose rate, bias voltage is 50 V.

Fig. R16 (Fig. S21) X-ray LoD of 4HPA detectors. (a) SNR values changes with incident dose rate, (b) LoD values changes with bias voltage s of the 4HPA detectors, (c) comparison of 4HPA detectors in this paper and other X-ray detectors of organic and perovskite single crystal devices in term of the S/J_{Dark} and LoD values.

Calculation method of signal-to-noise ratio (SNR)¹

The signal current (I_{signal}) was derived by subtracting the average photocurrent (I_{photo}) by the average dark current (I_{dark}). The noise current (I_{noise}) was obtained by calculating the standard deviation of the photocurrent.

$$I_{\text{signal}} = \bar{I}_{\text{photo}} - \bar{I}_{\text{dark}} \quad (\text{R5})$$

$$I_{\text{noise}} = \sqrt{\frac{1}{N} \sum_1^N (I_i - \bar{I}_{\text{photo}})^2} \quad (\text{R6})$$

Then the signal-to-noise ratio (SNR) was calculated as

$$\text{SNR} = I_{\text{signal}}/I_{\text{noise}} \quad (\text{R7})$$

4. I don't see authors claim of advantage of biocompatible since it should have a electrode, wire for the HV (low power) and preamp (ASIC) to avoid noise should be included for the clinical application even if 4HPA is a biocompatible.

Answer: We appreciate for the reviewer's comment. Biocompatible 4HPA detectors are important for both wearable and implantable personal dosimeter.

For wearable dosimeter, biocompatible sensor materials are better than those containing toxic elements, like Cd, Te and Pb, without requiring encapsulation layer, benefiting to thinner and more flexible devices that could be attached directly on human skin.

- For alpha particle detection, 50 μm -thick 4HPA detector can totally absorb the incident alpha particles, so we can decrease the thickness of 4HPA detectors to 50~100 μm . Then, the bias voltage can be decreased to 4 ~ 8 V (800 V cm^{-1}). For X-ray detection, around 100 μm -thick organic thick-film detectors are as effective as bulk single crystals, as we previously demonstrated in 4HCB case, low bias voltage (0.1 V) is possible¹⁸. We also can fabricate 4HPA thick film as the same way for flexible X-ray detectors with low bias voltage around 0.1 V.
- Therefore, such radiation sensor could be integrated with ultraflexible organic photovoltaic module as power supply¹⁹, small customized application specific integrated circuit (ASIC) chip for data collection and processing²⁰, and wireless communication module. The similar device has been demonstrated in the previous work by our group for photoplethysmogram (PPG) sensor, as shown in Fig. R17. Similar technology could be used for radiation sensor based on 4HPA detectors. We can use the 4HPA detector as the OPD part, **the whole device could be directly attached on human skin due to the biocompatibility of 4HPA, to achieve lightweight, flexible, and comfortable radiation monitoring sensor**. Then the radiation monitoring signal could be upload to cell phone or personal computer, achieving self-powered and wireless radiation monitoring.

Fig. R17 (a) Schematic diagram of the flexible, self-powered skin sensor on human hands, (b) Schematic diagram of the self-powered skin sensor with OPD and OPV module¹⁹.

For implantable application, the current method using an injected capsule including radiation detector, ASIC chips for data processing, and other supporting circuitry for power supply (0.1~8 V) would be very good for application²⁰. We can use the same method for 4HPA detector. Although the capsule should be sealed for avoid interaction between human body and devices, but the use of the biocompatible detector would be better for the safety consideration.

We revised main manuscript:

“For example, we can fabricate 4HPA thick film sensor with thickness around 100 μm as we demonstrated for 4HCB¹⁸. With good radiation detection performances and flexibility, such radiation sensor could be integrated with flexible organic photovoltaic module as power supply¹⁹, small customized application specific integrated circuit (ASIC) chip for data collection and processing²⁰, and wireless communication module. The whole device could be directly attached on human skin due to the biocompatibility of 4HPA, to achieve lightweight, flexible, and comfortable radiation monitoring sensor. Then the radiation monitoring signal could be upload to cell phone or personal computer, achieving self-powered and wireless radiation monitoring.” in Page 4 and 5.

5) For claimed application, low voltage is required. For example, Si sensor operate lower voltage and it is robust, much better performance.

Answer: We appreciate for the reviewer's comment. As we described in response for **Comment-4**. We can decrease the applied bias voltage if we make thinner 4HPA detectors or make interdigital electrodes.

For alpha particle detection, 50 μm -thick 4HPA detector can totally absorb the incident alpha particles, so we can decrease the thickness of 4HPA detectors to 50~100 μm . Then, the bias voltage can be decreased to 4 ~ 8 V (800 V cm^{-1}). For X-ray detection, around 100 μm -thick organic thick-film detectors are as effective as bulk single crystals, as we previously demonstrated in 4HCB case, low bias voltage (0.1 V) is possible¹⁸. We also can fabricate 4HPA thick film as the same way for radiation detectors.

For example, we fabricated 4HPA detectors on interdigital electrode patterned Polyethylene terephthalate (PET) substrate. We can also measure very good X-ray response at very low bias voltage (**Fig. R18**).

Fig. R18 (a) Picture of 4HPA film fabricated on interdigital electrode patterned PET substrate, (b) X-ray response of 4HPA film with comparison with 4HPA bulk single crystals.

6) For the clinical application ion-chamber and Si sensors are being used commonly authors need to compare the performance of 4HPA with that.

Answer: We appreciate for the reviewer's comment. The advantages and disadvantages for clinical applications in terms of radiation detection performance of commonly used radiation dosimeters including Thermoluminescent dosimeter (TLD), ion-chamber and Si were compared with 4HPA, as shown in Table. R3.

Table. R3 is added in Table. S2 in SI file.

For clinical applications, high detection performances, for example, high sensitivity, high energy resolution and fast response are very important. However, compact size, tissue equivalence, and biocompatibility of detector materials are also important.

Semiconductor-type dosimeters have higher signal transformation efficiency and compact size, thus benefiting to highly localized (or high spatial resolution) and easy-carry personal dosimeter. For example, Si and 4HPA.

In further, **tissue equivalence**, that means the effective atomic number (Z_{eff}) or density of the detector material is similar to the average human tissue Z (7.64 for muscles and density around 1.1 g cm^{-3}), is particularly important for personal dosimetry in radiotherapy and radiobiology. Only when the Z_{eff} or density of the dosimeter is matched to the value of human tissue can the dose value be obtained **without complex correction**. In this case, **4HPA detectors** (density of 1.25 g cm^{-3}) have much better **tissue equivalence** than **Silicon** (density of 2.33 g cm^{-3}). Furthermore, such tissue equivalent detectors with low attenuation efficiency but high sensitivity due to photoconductive gain can also be placed between the X-ray source and the patient, allowing a **highly localized, real-time radiation exposure monitoring**, which could not be achieved by inorganic semiconductors with large X-ray absorption rate.

For wearable dosimeter, **biocompatibility** and **flexibility** are very important factors. Si has limited biocompatible due to the toxic by-products during the fabrication process while solution grown 4HPA has superior biocompatibility. Even thin-film Si wafer is very brittle, less of flexibility, but 4HPA possessing two-dimensional structures has better mechanical bendable properties.

In conclusion, we don't want use 4HPA to compete with the state-of-the-art Si detectors, instead, we want to explore new possibilities for application using new materials. This work demonstrates that 4HPA detectors as the **world-first biocompatible and highly stable semiconductor-type radiation detectors** that can **directly achieve efficient X-ray, charged particles, and fast neutron detection**. Together with its compact, highly localized and tissue-equivalent properties, 4HPA detectors are very promising for radiation imager for **complementary X-ray (sensitive to heavy atoms) and fast neutrons (sensitive to light atoms) imaging** and **wearable/implantable personal dosimeters** for in-vivo and broad-band radiation monitoring during radiotherapy and other medical checking applications.

Table. R3. Advantages and disadvantages of current dosimeters and advances of biocompatible 4HPA organic detectors

Materials	Advantages	Disadvantages	Advance of 4HPA detectors
Thermoluminescent dosimeter (TLD)	 • Small size • Cheap • Available in various forms 	 • Not real-time • Need complex calibration 	 • Real-time • Energy-resolved • Tissue equivalent • Compact & small volume • Low-voltage supply (thick film) • High resistivity ($10^{12} \Omega \text{ cm}$) • Superior detection limit (20 nGy) • Insensitive to T and visible light • Direct fast neutron detection • Tissue-equivalent • Superior biocompatibility • light weight (1.25 g cm^{-3}) • Flexibility • Low-cost solution method
Ion chamber	 • Real-time • Precise 	 • Bulky size and visible • High voltage supply 	
Si	 • High carrier mobility • High energy resolution • Compact • High spatial resolution • Fast response 	 • Low resistivity • Poor detection limit ($\sim \text{mGy}$) • T-dependent response • No fast neutron detection • Non-tissue equivalent • Limited biocompatibility • Large density (2.33 g cm^{-3}) • Brittle thin-film wafer • High-cost fabrication 	

T means temperature here.

7) Authors claim wrote “normally contains a fast-rising (below 1 μ s) and an accompanied slow-rising components (tens of microseconds) in organic detectors while that usually below 1 μ s in inorganic detectors, e.g., CdZnTe 15,17. Whether seeking novel materials or possible solutions to improve the charge transport in OSCS” It is not inorganic semiconductor property but people are using the semiconductor which has electron and hole mobility are similar such as CZT, CT, Si, HPGGe. If either electron or hole mobility is very low like 4HCB or 4HPA, it is difficult to use for spectroscopic application since there is slow rising time which depends on the position. Thus, authors need to explain how 4HPA can get good performance when hole mobility is much lower than electron mobility.

Answer: We appreciate for the reviewer’s comment.

To the best of our knowledge, even CdZnTe (CZT), CdTe (CT) have lower hole mobility ($80\sim 30\text{ cm}^2\text{ V}^{-1}\text{ s}^{-1}$) than their electron mobility ($1000\sim 800\text{ cm}^2\text{ V}^{-1}\text{ s}^{-1}$)²¹. Actually, we don’t need to consider the difference between electrons’ and holes’ mobilities for alpha particles detection due to the shallow penetration depth of alpha particles. This problem is significant for γ -ray detection, however, **single polarity charge sensing technique** (output signal almost only from single-type carriers) is commonly utilized in conventional semiconductor radiation detectors (CdZnTe and CdTe) to avoid the significant trapping effects²². This is similar to our case in 4HPA, therefore, we can use many methods that are developed to achieve the good performances in CdZnTe and CdTe. Detailed explanation as following:

For **alpha particle detection**, the penetration depth is normally below than 50 μ m that is much smaller compared with the thickness or the charge carrier drift length (\sim mm). Therefore, the good performance can be obtained by **single-carrier collection**. Alpha particles are irradiated from cathode side and the ionized electrons are collected from anode. The ionized holes will get recombination very soon while electrons can transport to the anode, generating electrical signal. In the 4HPA detectors, electron’s mobility is $4.17\pm 0.040\text{ cm}^2\cdot\text{V}^{-1}\cdot\text{s}^{-1}$ while the hole’s mobility is $0.06\pm 0.001\text{ cm}^2\cdot\text{V}^{-1}\cdot\text{s}^{-1}$, so we use the same method as CZT crystals, as shown in **Fig. R19**.

Fig. R19 (a) Schematic diagram of alpha particles detection by detectors that electrons have larger mobility than holes.

For **spectra detection of γ -ray**, the penetration depth is very deep, thus the electro-hole pairs can be generated within the whole detector thickness, and moving of these electrons and holes will induce the charge on the electrode Q and output signal. According to Shockley-Ramo theorem, the induced charge on the electrode Q by a moving point charge q are given by:

$$Q = -q\varphi_0(\mathbf{x}) \quad \text{R8}$$

where $\varphi_0(\mathbf{x})$ is the electric potential that would exist at q 's instantaneous position \mathbf{x} , and is called the weighting potential.

For conventional devices using planar electrodes, the $\varphi_0(\mathbf{x})$ is simple a linear function of depth \mathbf{x} . In this case, the output signal is generated by the movement of both electrons and holes. However, in this case, charge trapping commonly occurs in even for typical CdTe, CdZnTe and HgI, that means the output signal depends not only on the deposited energy of γ -ray, but also on the position of that interaction. Therefore, the deposited energy cannot be obtained uniquely from the output signal. In order to overcome the trapping effect, **single polarity charge sensing technique** is used for the CdTe and CdZnTe²². This is similar condition of 4HPA detectors case, in which holes are trapped and have lower mobility while the electrons have better transport properties. Therefore, the optimized methods for high detection performance of CdZnTe/CdTe are applicable for our 4HPA case. For example, if we change the planar electrode structure to **coplanar grid electrode**, the weighting potential will also be changed, as shown in **Fig. R20**²². The **single polarity charge sensing** by electrons can be achieved because the holes move in the linear region of weighting potentials ($0 \sim 1-P$) and electrons are collected by anode (#2).

We revised main manuscript:

“It worth note that the large difference of electron and hole mobility will not influence the spectra detection of alpha particles if alpha particles are irradiated from cathode side and the ionized electrons are collected from anode.” in Page 11.

Fig. R20 Electrodes structure design method for γ -ray detection by CdZnTe and CdTe: Electrode structures and weighting potentials of (a) and (b) planar detectors, (c) and (d) coplanar grid electrodes for electrons signal detection²².

8) Authors need to show fitting results of alpha particle energy resolution. It is not Gaussian and it shows a shoulder and tail, why? See Ref 16 for the comparison.

Answer: We appreciate for the reviewer’s comment. The fitting result of alpha particle energy resolution is shown.

The spectra based on 4HPA detectors are not typical gaussian. In the range of low channels before the peak, e.g., 0~50 channels for bias voltage of 1000 V along the a axes of 4HPA detectors, the channels for dark noise and the part of channels for signal peak was mixed together, which may come from the measurement system noise, e.g., the incompatibility between the organic semiconductors and the electric circuit that is designed for conventional inorganic semiconductors, which also influence the tail effect for 4HPA detectors. In addition, due to the large bandgap of 4HPA (3.8 eV), the number of ionized electrons is less than other inorganic detectors (e.g., CdTe, $E_g=1.55$ eV), so

the signal amplitude obtained by 4HPA detector is smaller than the signal from inorganic semiconductors. Therefore, the signal and the noise channels are very close for the case of 4HPA detectors. Both two reasons will lead to partially mixed signal and noise. Therefore, for energy resolution fitting, we selected the Gaussian part of the spectra. For example, **Fig. R21** shows the Gaussian part of the spectrum obtained by the 4HPA detectors along the *a* axes. For this part, we utilized the Gaussian fitting and the result was automatically given by the Origin software, obtaining energy resolution of 35% when bias voltage is 1000 V.

Fig. R21 Gaussian part of alpha particle spectrum obtained by 4HPA detector and fitting result for energy resolution around 35%.

9) Authors should not use not well-defined quantity such as “still requires a long duration time around 30 minutes for a spectrum measurement” or 0.5-10 sec in Fig 5 since does rate is not known.

Answer: We appreciate for the reviewer’s comment. In this paper, we used ^{241}Am as alpha particle source for 4HPA detectors, with incident alpha particle activity (A) in the order of 10^6 Bq. We used the similar alpha source 4HCB in our previous work, with the $A \sim 10^5$ Bq²³.

We revised main manuscript:

“**Fig. 5(e)-(k)** give the alpha particle spectra with detection time from 0.04 s to 60 s at bias voltage of 200 V, the incident alpha particle activity (A) in the order of 10^6 Bq, which is around 10 time higher than alpha source for 4HCB²³.” in Page 12.

10) Authors claimed as “4HPA detectors exhibit the record spectra detection of α particles among their organic counterparts, with energy resolution of 36%, (μt) e product of $(4.91 \pm 0.07) \times 10^{-5} \text{ cm}^2 \text{ V}^{-1}$, and spectra detection time down to 3 ms” As I mentioned spectral detection time down to 3ms is not meaningful unless the total alpha rate is described. (with a few alphas, Si can detect it with good energy resolution) According to Ref 21, $(\mu\text{t})\text{h} = 8.5 \times 10^{-5} \text{ cm}^2/\text{V}$ which is better than authors results. Even if they don’t give an energy resolution of an alpha particle, it looks better than authors results at HV.

Answer: We appreciate for the reviewer’s comment. In this paper, we used ^{241}Am as alpha particle source, with incident alpha particle activity (A) of around 10^6 Bq. The irradiated alpha source was similar with 4HCB case ($A \sim 10^5$ Bq) in Ref²³.

In previously reported paper for 4HCB organic detectors, Ref²³ and Ref²⁴, the digital pulse shaper was used instead of the amplifier (ORTEC, 572A) for 4HPA detectors and most of other radiation detectors including CdTe, CdZnTe and perovskites. Therefore, the obtained μt values by these two-measurement system may not be compared directly.

By **digital pulse shaper** (simulated-type amplifier), we can set **filter** and optimize the measurement parameters to cut down the noise and enhance the signal-to-noise ratio,

but the counts are much lower than ORTEC 572A amplifier due to the designed filter process. Therefore, the counts of alpha-particle spectra of 4HCB (collection time of 30 min, alpha source $A \sim 10^5$ Bq, bias voltage of 600 V) are around **120 times lower** than that 4HPA (collection time of 1 min, alpha source $A \sim 10^6$ Bq, bias voltage of 600 V) (**Fig. R22**), which makes detection time for an energy spectrum is much longer (30 min) if using 4HCB detectors.

Fig. R22 Counting comparison of alpha particle spectra obtained by (a) 4HCB detectors using digital pulse shaper (collection time of 30 min, alpha source $A \sim 10^5$ Bq) and (b) 4HPA detectors using ORTEC 572A amplifier (collection time of 1 min, alpha source $A \sim 10^6$ Bq).

11) Geant4 is not the best tool for alpha particle simulation, SRIM is recommended in this application.

Answer: We appreciate for the reviewer's comment.

We compared the results from Geant4 and SRIM simulation, the measured penetration depth values by these two methods are almost same, 35.2 μm for Geant4 simulation (**Fig. R23(a)**) and 35.6 μm by SRIM simulation (**Fig. R23(b)**).

We revised the **Fig. S11**.

“According to calculation by both Geant4 and SRIM software, the effective thickness of 4HPA semiconductors for charged particles detection is around 35 μm (**Fig. S11**).”

Fig. R23 (Fig. S11) Penetration depth of 5.49 MeV alpha particles from ^{241}Am source into 4HPA single crystal detectors simulated by (a) Geant4 software and (b) SRIM software.

SRIM would be better software for simulation of the penetration depth of alpha particles into 4HPA detectors, but it could not be used for simulating the energy distribution of the alpha-particles ionized electrons. Therefore, we use Geant4 here for simulating the penetration depth of alpha particles and the energy distribution spectra of ionized electrons, as shown in **Fig. R24**. The simulation is conducted by Geant4 package using C++ toolkit, which can probe the interaction between materials and incident particles²⁵. For the Geant4 simulation, the geometry (G4VUserDetectorConstruction) is defined as 4HPA, $\text{C}_8\text{H}_8\text{O}_3$, density of 1.253 g cm^{-3} , size of $6 \times 6 \times 10 \text{ cm}^3$; the particle beam source (G4VPrimaryGeneratorAction) is defined as linear α beam with diameter of 2 mm, energy of 5.49 MeV, penetrating direction perpendicular to the 4HPA surface; the physical interaction (G4VUserPhysicsLists) is FTFP_BERT. The results are the initial energy, penetrating depth, and energy deposit for each particle-material interaction.

Fig. R24 Energy distribution spectra of electrons ionized by 5.49 MeV α particles. (a) alpha particles ionized electrons, (b) second ionized electrons, (c) multiple ionized electrons.

12) In the clinical application with X-ray or gamma-rays, it can not be used for the spectroscopic mode since intensity is high, thus authors need to also show integration mode with X-ray.

Answer: We thank for the reviewer's comment. We have added the X-ray detection methods and results by integrated detection mode in main manuscript and SI files.

Materials and Methods:

“X-ray detection and imaging

For X-ray detection, we used Amptek Mini-X tube to generate X-ray beam. During the experiment, a 0.15-mm thick Cu attenuator was used to weaken the incident X-ray dose rate, the distance between the X-ray source and 4HPA detectors were 10 cm. The X-ray source was operated with tube voltage of 40 kVp (X-ray photo energy up to 40 keV and peak intensity at 27 keV) and adjustable tube current in range of 0.005-0.06 mA to control the dose rate of 0.35~2.34 $\mu\text{Gy}_{\text{air}} \text{ s}^{-1}$. For X-ray irradiation, Spellman X4060 X-ray generator was used. The largest dose rate of 19.18 $\text{mGy}_{\text{air}} \text{ s}^{-1}$ was obtained with the tube voltage of 150 kVp and the tube current of 1.2 mA. The total irradiation time is a continuous 10 hours for each 4HPA detectors. The X-ray dose rate was carefully calibrated by using Fluke Si diode (RaySafe X2) dosimeter. The Keithley 6517B was utilized here for applying the high bias voltage and measuring the dark and photocurrent with high precision. All the measurements were conducted in air condition and room temperature.”

Main manuscript:

“2.6 X-ray Detection and Imaging by 4HPA Detectors Using integration mode.

In the clinical application, X-ray detection with integration mode is also important, which is demonstrated by our 4HPA detectors along the *a* axis.

Fig. 6(a) shows the typical Photocurrent-Time (**I-t**) curve of 4HPA detectors, with a dose rate range of 0.35~2.34 $\mu\text{Gy}_{\text{air}} \text{ s}^{-1}$. The bias voltages of 5~130 V are applied along the *a* axis of 4HPA single crystals. No hysteresis or appreciable current drift is observed, which is comparable with other highly performed detectors²⁶⁻³⁰. The dark current is stable after repeated exposure under X-ray beam, with a value of 0.1 pA at bias voltage of 150 V and a very low dark current drift $\sim 10^{-9} \text{ nA cm}^{-1} \text{ s}^{-1} \text{ V}^{-1}$ (**Fig. S19**). It is noteworthy that the stable and repeatable detectable dose rate 0.35 $\mu\text{Gy}_{\text{air}} \text{ s}^{-1}$ has been experimentally achieved (**Fig. S20**). The **Signal-to-Noise (SNR)** ratio of photocurrent under each dose rate with each bias voltage were calculated (**SI 3**), by

which the **Detection of Limit (LoD)** values at each bias voltage (**Fig. 6(b)**) were estimated according to the SNR value larger than IUPAC⁸ defined 3. The best **LoD** of $20 \text{ nGy}_{\text{air}} \text{ s}^{-1}$ (275 times lower than required $5.5 \text{ } \mu\text{Gy}_{\text{air}} \text{ s}^{-1}$ for regular medical diagnostics⁹).

Then the linear Photocurrent density and Dose rate (**J-D**) relation was shown in **Fig. 6(c)**, by which the **Sensitivity (S)** was calculated using linear fitting, with **S** of $330 \text{ } \mu\text{C Gy}_{\text{air}}^{-1} \text{ cm}^{-2}$ at a bias voltage of 150 V or $16612 \text{ } \mu\text{C Gy}_{\text{abs}}^{-1} \text{ cm}^{-3}$ when considered the X-ray absorption rate by 4HPA detectors ((**Fig. 6(d)**), **SI 4**). Then, the X-ray detection **Sensitivity (S)/Dark current density (J_{Dark})**⁵ as high as $1.5 \times 10^{14} \text{ } \mu\text{Gy}_{\text{air}}^{-1} \text{ s}$ was achieved by 4HPA detectors for 40 kVp X-ray detection (**SI 5**). In general, 4HPA show superior X-ray detection performances using integrated mode, which even comparable with the state-of-the-art halide perovskite X-ray detectors (**Fig. 6(e)**)^{5,10-17}. The good detection performance can be achieved in 4HPA organic detectors with weak X-ray absorption rate (**SI 4**) thanks to the high photoconductive gain or **Charge Collection Efficiency (CCE)** up to 15000% (**SI 4**), which may originate from the charge injection induced by the surface layer edge states on 4HPA.

Both the low dark current and highly stable SNR results in a very low level of baseline drift and highly stable photocurrent output signal. Therefore, the 4HPA detector exhibit superior direct imaging capability, **Fig. 6(f-g)**. The high colour contrast demonstrates that the object is clearly resolved, furthermore, it shows unambiguously the similarity in shape and size between the nut and the X-ray image which means that there is no variation or deviation. The as-fabricated 4HCB devices also show good stability after long term operation, which is vital for the actual imaging application (**Fig. 6(h-i)**, **SI 6**.”

Fig. R25 (Fig. 6) X-ray detection and imaging performance of 4HPA detectors. (a) I-T curves with decreased X-ray dose rate and switching X-ray “ON/OFF” at bias voltage from 5~130 V, (d) LoD changes with bias voltage, (c) photocurrent density (J) changes with irradiated X-ray dose rate at bias voltage from 5~150 V, (d) **Sensitivity** of 4HPA detectors for 40 kVp X-ray at a series of bias voltage, (e) detection performance comparison of 4HPA and other organic and halide perovskite X-ray detectors in terms of the S/J_{Dark} and the **LoD** values, (f) metal wire in plastic capsule and (g) its X-ray imaging by 4HPA imager detectors, (h) long-term work stability and (i) continuous 10-hour X-ray irradiation (150 kVp, total dose 690.48 Gy_{air}) stability.

Reviewer #2 (Remarks to the Author):

The spectra detection of charged particles exhibiting single-particle sensitivity is common in the realm of particle detection. Achieving 100% efficiency in detecting charged particles is always possible; the pivotal factor lies in the effective collection of the particle's charge, i.e., CCE, which directly impacts energy resolution. Silicon (Si) detectors routinely attain an energy resolution as fine as 1%. The reported energy resolution of 36% by this work is unimpressive. This remarkable of this work is attributed to the device's flexible format for wearable devices, which holds a distinctiveness worthy of publication. BTW, the usage of the term "energy FWHM 36%" is inaccurate. Equally remarkable is the measured resistivity of $(1.28 \pm 0.003) \times 10^{12} \Omega \cdot \text{cm}$ from this work.

Answer: We thank for the reviewer's comment. We have revised the manuscript according to the comments.

1. "For the first time, the charge transport mechanism responsible for the superior performance is clarified in view of high-energy physics".

I was not quite understood this statement. Firstly I would never claim "for the first time", we don't know what we don't know. What has the high energy physics to do with this device? What energy range is considered the domain for high energy physics? To my knowledge, it is the order of magnitude of the energy dealt with by this paper.

Answer: We appreciate for the reviewer's comment. We revised this sentence in abstract to "The two-dimensional anisotropic charge transport mechanism responsible for the superior performance is clarified." in Page 1.

2. “Geant4 simulation indicates that 5.49 MeV α particles 154 can ionize electrons with energy of 0-150 eV (Fig. S9),”

How is zero energy possible for ionized energy?

Answer: We appreciate for the reviewer’s comment. The accurate energy range is 0.8 ~ 150 eV, as shown in **Fig. R26**.

We revised this sentence in main manuscript:

“In addition, **Fig. 3(d)** and **(e)** give the Monte Carlo simulated energy spectra of α particles excited electrons after first ionization and multiple ionization, respectively, indicating that radiation generated electrons possess energy of 0.8~150 eV before the thermal relaxation.” in Page 6.

Fig. R26 Energy distribution spectra of electrons ionized by 5.49 MeV α particles in Fig. S9 with (a) Linear axes, (b) log-log axes.

3. Fig 3c should be just a Bragg peak, but it doesn't seem like the SRIM/TRIM simulation of the stopping power curve. I would double-check and verify Geant 4 simulation.

Answer: We appreciate for the reviewer's comment. Fig. 3c is the energy deposition of 5.49 MeV α particles in 4HPA detectors changes with detector thickness simulated by **Geant4** software. Same result as shown **Fig. S10** in previous **SI file (Fig. R27)**. We use Geant4 software for simulation of alpha particle penetration depth into 4HPA detectors and the energy distribution spectra of alpha particles ionized electrons in 4HPA detectors because **SRIM** could not calculate the energy of ionized electrons. We have compared the alpha particle penetration results obtained by **Geant4** and **SRIM**, which are almost same, 35.2 μm for Geant4 measurement **Fig. R28(a)**, 35.6 μm by SRIM simulation **Fig. R28(b)**.

We added the following description in **Fig. S11**.

“According to calculation by both Geant4 and SRIM software, the effective thickness of 4HPA semiconductors for charged particles detection is around 35 μm (**Fig. S11**).”

Fig. R27 (Fig. S10) Energy deposition of 5.49 MeV α particles in previous Fig. S10. (a) Energy deposition in 4HPA detectors changes with sample thickness, (b) the energy of incident α particles changes with penetration depth.

Fig. R28 (Fig. S11) Energy deposition of 5.49 MeV α particles in 4HPA detectors simulated by (a) Geant4 software, and (b) SRIM software.

4. “The 4HPA detector was positioned inside a copper chamber at a distance of 1 cm from the ^{241}Am alpha particle source. “

Is the copper chamber under vacuum? Are there any wire near the radiation field, in other words, how do you rule out the signal is not coming from air ionization?

Answer: We appreciate for the reviewer’s comment. We added the following description in **Materials and Methods**.

“During the measurement process, we shielded the whole device using a paper with thickness of 2 mm, excepting for a hole (diameter of 3 mm) to irradiate the alpha particles to cathode or anode of 4HPA detectors to avoid any possible ionization from the other objects or wire.”

As for the air ionization, the copper chamber is not in the vacuum, and we also use the same experimental set-up for CdZnTe and halide perovskite detectors^{31,32}. But we compare the spectra without and with 4HPA detectors measured by same equipment and in same environment (air and room temperature), as shown in **Fig. R29**.

Fig. R29 Comparison of spectra without and with 4HPA detectors when the bias voltage is 1000 V.

Reviewer #3 (Remarks to the Author):

This paper reports the solution-grown biocompatible n-type organic single-crystalline semiconductor materials for real-time spectra detection of charged particles with single-particle sensitivity, finding a “heavy-to-light electron transition” dominated charge-transport mechanism for organic radiation detectors. This novel radiation detector possesses excellent biocompatibility (cell viability over 90% after 24-hour incubation concentration of 2 mg ml⁻¹), which is comparable to the state-of-the-art carbon-based biocompatible materials. In the meanwhile, the detector achieves recorded charge-particle detection performance within their organic counterparts, with energy resolution of 36%, spectra detection time down to 3 ms, and mobility-lifetime product of electrons of $(4.91 \pm 0.07) \times 10^{-5} \text{ cm}^2 \text{ V}^{-1}$. Furthermore, the author combines DFT, Monte Carlo simulation, and time-dependent electrical measurements, proposing a new charge transport mechanism for organic radiation detectors, which is very interesting and significant to understand the charge-transport behaviours of organic semiconductors under irradiation and design new organic radiation detectors.

Overall, this paper demonstrates the first OSCS detector as low-cost consumer electronics for wearable/implantable dosimeters with real-time, position-sensitive, and in-vivo healthcare monitoring in radiation-exposure environments; and broadens the charge-transport theory of organic semiconductor from the viewpoint of high-energy physics. Therefore, I recommend this paper to publish on Nature Communications after addressing following comments.

Answer: We thank for the reviewer’s positive comment. We have revised the manuscript according to the comments.

1. What are conventional detector materials for in-vivo radiation monitoring?
Compared with the conventional material, the benefits to use 4HPA organic detectors should be described.

Answer: We appreciate for the reviewer's comment. For in-vivo application, Metal-Oxide Semiconductor Field Effect Transistor (MOSFET) dosimeters based on Si transistors are commonly utilized. The benefits of 4HPA compared with Si MOSFET dosimeters and other conventional dosimeters for clinical applications are shown in Table. R4.

Table. R4 is added in Table. S2 in SI file.

For clinical applications, high detection performances, for example, high sensitivity, high energy resolution and fast response are very important. However, **compact size**, **tissue equivalence**, and **biocompatibility** of detector materials are also important.

Semiconductor-type dosimeters have higher signal transformation efficiency and compact size, thus benefiting to highly localized (or high spatial resolution) and easy-carry personal dosimeter. For example, Si and 4HPA.

In further, **tissue equivalence**, that means the effective atomic number (Z_{eff}) or density of the detector material is similar to the average human tissue Z (7.64 for muscles and density around 1.1 g cm^{-3}), is particularly important for personal dosimetry in radiotherapy and radiobiology. Only when the Z_{eff} or density of the dosimeter is matched to the value of human tissue can the dose value be obtained **without complex correction**. In this case, **4HPA detectors** (density of 1.25 g cm^{-3}) have much better **tissue equivalence** than **Silicon** (density of 2.33 g cm^{-3}). Furthermore, such tissue equivalent detectors with low attenuation efficiency but high sensitivity due to photoconductive gain can also be placed between the X-ray source and the patient, allowing a **highly localized, real-time radiation exposure monitoring**, which could not be achieved by inorganic semiconductors with large X-ray absorption rate.

For wearable dosimeter, **biocompatibility** and **flexibility** are very important factors. Si has limited biocompatible due to the toxic by-products during the fabrication process while solution grown 4HPA has superior biocompatibility. Even thin-film Si wafer is

very brittle, less of flexibility, but 4HPA possessing two-dimensional structures has better mechanical bendable properties.

This work demonstrates that 4HPA detectors as the **world-first biocompatible and highly stable semiconductor-type radiation detectors** that can **directly achieve efficient X-ray, charged particles, and fast neutron detection**. Together with its compact, highly localized and tissue-equivalent properties, 4HPA detectors are very promising for radiation imager for **complementary X-ray (sensitive to heavy atoms) and fast neutrons (sensitive to light atoms) imaging** and **wearable/implantable personal dosimeters** for in-vivo and broad-band radiation monitoring during radiotherapy and other medical checking applications.

Table. R4. Advantages and disadvantages of current dosimeters and advances of biocompatible 4HPA organic detectors

Materials	Advantages	Disadvantages	Advance of 4HPA detectors
Thermoluminescent dosimeter (TLD)	 • Small size • Cheap • Available in various forms 	 • Not real-time • Need complex calibration 	 • Real-time • Energy-resolved • Tissue equivalent • Compact & small volume • Low-voltage supply (thick film) • High resistivity ($10^{12} \Omega \text{ cm}$) • Superior detection limit (20 nGy) • Insensitive to T and visible light • Direct fast neutron detection • Tissue-equivalent • Superior biocompatibility • light weight (1.25 g cm^{-3}) • Flexibility • Low-cost solution method
Ion chamber	 • Real-time • Precise 	 • Bulky size and visible • High voltage supply 	
Si	 • High carrier mobility • High energy resolution • Compact • High spatial resolution • Fast response 	 • Low resistivity • Poor detection limit ($\sim \text{mGy}$) • T-dependent response • No fast neutron detection • Non-tissue equivalent • Limited biocompatibility • Large density (2.33 g cm^{-3}) • Brittle thin-film wafer • High-cost fabrication 	

T means temperature here.

2. The 4HPA detectors show better detection performance and biocompatibility than the reported 4HCB detectors. What is the difference of 4HPA and 4HCB molecules? How does it influence the physical properties of these two materials?

Answer: We appreciate for the reviewer's comment. We added following description in **Fig. S3**.

“4HPA and 4HCB molecules only have one different function group, the -CN group in 4HCB while -CH₂COOH group in 4HPA. Compared with -CN group, -CH₂COOH group results in better biocompatibility of 4HPA (**Fig. S3(a-b)**). In addition, the difference of the functional group in 4HCB and 4HPA molecules also induces similar packing type but different periodical properties in single crystals (**Fig. S3(c-d)**). For example, 4HPA has similar quasi-two-dimensional crystal structures as 4HCB, with intermolecular bonds along the *a* and *b* axes mainly rely on the $\pi - \pi$ bonds while that of the *c* axis is hydrogen bonds. However, compared with 4HCB, the distance (unit cell along main crystal axes: *a* = 0.5 nm, *b* = 0.9 nm, *c* = 1.5 nm) between two neighbouring molecular with $\pi - \pi$ bonds and hydrogen bonds are smaller than 4HCB (unit cell along main crystal axes: *a* = 0.9 nm, *b* = 1.0 nm, *c* = 2.5 nm). This indicates that the charge transport properties of 4HPA is better than 4HCB due to higher-degree of $\pi - \pi$ overlaps of neighbouring molecules.”

Fig. R30 (Fig. S3) Molecular structures of 4HCB and 4HPA. Single molecular structures of (a) 4HCB and (b) 4HPA, molecular packing of (c) 4HCB and (d) 4HPA.

3. In this paper, the author reports a very simple solution method for 4HPA single crystal growth. Since the crystal quality is very important for detection performance, how does the author control the quality of 4HPA single crystals?

Answer: We appreciate for the reviewer's comment. Here, we utilized slow solvent evaporation method to growth 4HPA single crystals with the size up to $10 \times 5 \times 2 \text{ mm}^3$. The low temperature ($0 \sim 5 \text{ }^\circ\text{C}$) and controlled solvent evaporation rate were utilized for 4HPA growth, detailed control method has been reported in our previous paper for 4HCB organic single crystals²³, as shown in **Fig. R31**.

We added the following description in **Fig. S1**.

“The low temperature ($0 \sim 5 \text{ }^\circ\text{C}$) and controlled solvent evaporation rate were utilized for 4HPA growth, detailed control method has been reported in our previous paper for 4HCB organic single crystals²³.”

Fig. R31 Schematic diagram of 4HPA crystal growth control.

4. What is the effective thickness of 4HPA semiconductors for charged particles detection? Is it possible to fabricate flexible 4HPA detectors for wearable/implanted devices?

Answer: We appreciate for the reviewer's comment. We added the following description in **Fig. S11**.

“According to calculation by both Geant4 and SRIM software, the effective thickness of 4HPA semiconductors for charged particles detection is around 35 μm (**Fig. S11**).”

Fig. R32 (Fig. S11) Energy deposition of 5.49 MeV α particles in 4HPA detectors simulated by (a) Geant4 software, and (b) SRIM software.

Possibility of flexible 4HPA sensors:

At present, the fabrication of flexible 4HPA semiconductors with thickness around 100 μm are under preparation by hot-press and drop casting methods. The hot-press method is similar to our previous work of 4HCB film¹⁸, the device composed of 100- μm thick 4HCB film/interdigital Au electrode/175- μm thick PET substrate exhibited very good performance for X-ray (sensitivity of 93 $\mu\text{C Gy}_{\text{air}}^{-1} \text{cm}^{-2}$) and charged particles detection with maintaining bending radius around 2 mm. Due to the similar structural and melting properties of 4HCB and 4HPA, this method is also adaptable to fabricate flexible 4HPA.

In addition, we can fabricate 4HPA detectors on interdigital electrode patterned Polyethylene terephthalate (PET) substrate by solution drop casting method. We can also measure very good X-ray response at very low bias voltage (**Fig. R33**).

Fig. R33 (a) Picture of 4HPA film fabricated on interdigital electrode patterned PET substrate, (b) X-ray response of 4HPA film with comparison with 4HPA bulk single crystals.

We revised main manuscript:

“For example, we can fabricate 4HPA thick film sensor with thickness around 100 μm as we demonstrated for 4HCB¹⁸. With good radiation detection performances and flexibility, such radiation sensor could be integrated with flexible organic photovoltaic module as power supply¹⁹, small customized application specific integrated circuit (ASIC) chip for data collection and processing²⁰, and wireless communication module. The whole device could be directly attached on human skin due to the biocompatibility of 4HPA, to achieve lightweight, flexible, and comfortable radiation monitoring sensor. Then the radiation monitoring signal could be upload to cell phone or personal computer, achieving self-powered and wireless radiation monitoring.” in Page 4 and 5.

5. How about the work stability of 4HPA detectors?

Answer: We appreciate for the reviewer’s comment. We have evaluated the work stability of 4HPA detectors in terms of integrated-mode (for detection of high-flux of irradiations) under high bias voltage and strong irradiation field. The following content was added in main manuscript and **SI 6**.

Main manuscript:

“The as-fabricated 4HCB devices also show good stability after long term operation, which is vital for the actual imaging application (**Fig. 6(h-i)** and **SI 6**.” in page 13.

“SI 6 Degradation with radiations

At first, we evaluated the long-time **I-t** curves with “ON/OFF” switching behaviors of 4HPA detectors. With the increasing X-ray dose rate, as shown in **Fig. S26(a)**, after 50 cycles (continuous operation of 3100 s), the 4HPA detectors didn’t show any degradation. Then, we measured the 100 “ON/OFF” switching cycles (continuous operation of 6000 s) under constant X-ray irradiation with dose rate of $19.18 \text{ mGy}_{\text{air}} \text{ s}^{-1}$ and bias voltage of 150 V. No degradation happens in both dark and photocurrent (**Fig. S26(b)**).

In addition, long-time current drift stability measurements of 4HPA devices were also carried out with continuous 2-hour dark current measurement and photocurrent measurement with X-ray irradiation (dose rate of $19.18 \text{ mGy}_{\text{air}} \text{ s}^{-1}$, total dose of $138.096 \text{ Gy}_{\text{air}}$). The result shows both the dark current drift (around $10^{-9} \text{ nA cm}^{-1} \text{ s}^{-1} \text{ V}^{-1}$) and photocurrent drift are very small, and no degradation occurs after continuous 2-hour device work period (**Fig. S27**).

In further, we utilized 150 kVp X-ray (dose rate of $19.18 \text{ mGy}_{\text{air}} \text{ s}^{-1}$) to irradiate 4HPA detectors for 10 hours, with total irradiation dose of $690.48 \text{ Gy}_{\text{air}}$ (upper limit of our X-ray generator). We compared the photodetection properties of before and after 10-hour irradiation, as shown in **Fig. S28**. Both dark current and photocurrent didn’t show any obvious degradation.

These results indicate 4HPA detectors show very good radiation stability and long-term work stability at high electric field.”

Fig. R34 (Fig. S26) X-ray detection long-term work stability. (a) X-ray photocurrent I-T cycles change with increased dose rate, (b) X-ray photocurrent I-T cycles when the work time continues to 6000 s with bias voltage of 150 V.

Fig. R35 (Fig. S27) X-ray detection long-term work stability of 4HPA detectors with continue 120 min work for X-ray detection.

Fig. R36 (Fig. S28) X-ray detection long-term work stability after continuously detection of 0 h, 9 h, 1 h, and 10 h and 150 kVp X-ray dose rate of $19180 \mu\text{Gy}_{\text{air}} \text{ s}^{-1}$, total irradiation dose of $690.48 \text{ Gy}_{\text{air}}$.

6. In Fig. S15, the author also gives the result of direct fast neutron detection using 4HPA detectors. Why does the 4HPA OSCS can detect X-rays and fast neutrons? More comments should be given on the potential applications of this properties.

Answer: We appreciate for the reviewer’s comment. The mechanism of 4HPA OSCS for X-ray detection is photoelectric effect and Compton scattering, in which the interaction cross-section is positively related to atomic number of 4HPA.

For direct fast neutron detection, we added the following description in **Fig. S17**.

“The direct detection of fast neutrons by 4HPA OSCS is due to 4HPA possesses high-density of H atoms ($\sim 2 \times 10^{22} \text{ n cm}^{-3}$) that have largest interaction possibility with fast neutrons to produce charged particles, and at the same time, 4HPA also can detect charged particles as a semiconductor detector, therefore achieving direct detection of fast neutrons (**Fig. S17**), as we described in Ref³³. It worth note that organic detectors like 4HPA OSCS has very high detection efficiency for direct fast neutron detection while most of inorganic semiconductors are not sensitive to fast neutrons.”

Fig. R37 (Fig. S17) Schematic diagram of direct fast neutron detection by 4HPA OSCS³³.

Potential applications: we added the following description in **main manuscript**.

“This study demonstrates that 4HPA detectors as the world-first biocompatible and highly stable semiconductor-type radiation detectors that can directly achieve efficient X-ray, charged particles, and fast neutron detection. Together with its compact, highly localized and tissue-equivalent properties, 4HPA detectors are very promising for radiation imager for **complementary X-ray (sensitive to heavy atoms) and fast neutrons (sensitive to light atoms) imaging and wearable/implantable personal dosimeters** for in-vivo and broad-band radiation monitoring during radiotherapy and other medical checking applications.” in page 14.

References

- 1 Pan, W. *et al.* Cs₂AgBiBr₆ single-crystal X-ray detectors with a low detection limit. *Nat. Photonics*. **11**, 726-732, doi:10.1038/s41566-017-0012-4 (2017).
- 2 Berger, M. J. e. a. XCOM: Photon Cross Sections Database: NIST Standard Reference Database 8 (NIST, 2013). <https://www.nist.gov/pml/xcom-photon-cross-sections-database>.
- 3 Alig, R. C. & Bloom, S. Electron-Hole-Pair Creation Energies in Semiconductors. *Phys. Rev. Lett.* **35**, 1522-1525, doi:10.1103/PhysRevLett.35.1522 (1975).
- 4 Zhao, D. *et al.* Photoconductive gain under low-flux X-ray irradiation in 4HCB organic single crystal detectors. *Appl. Phys. Express*. **13**, 071004, doi:10.35848/1882-0786/ab9adb (2020).
- 5 Jin, P. *et al.* Realizing nearly-zero dark current and ultrahigh signal-to-noise ratio perovskite X-ray detector and image array by dark-current-shunting strategy. *Nat. Commun.* **14**, doi:10.1038/s41467-023-36313-6 (2023).

- 6 Nan, R., Jie, W., Zha, G. & Yu, H. Relationship between high resistivity and the deep level defects in CZT: In. *Nucl. Instrum. Methods. Phys. Res. A* **705**, 32-35 (2013).
- 7 Mandal, K. C., Krishna, R. M., Pak, R. O. & Mannan, M. A. in *Hard X-Ray, Gamma-Ray, and Neutron Detector Physics XVI*. 278-286 (SPIE).
- 8 Thompson, M., Ellison, S. L. & Wood, R. Harmonized guidelines for single-laboratory validation of methods of analysis (IUPAC Technical Report). *Pure Appl. Chem.* **74**, 835-855 (2002).
- 9 Clairand, I. *et al.* Use of active personal dosimeters in interventional radiology and cardiology: Tests in laboratory conditions and recommendations - ORAMED project. *Radiation Measurements* **46**, 1252-1257, doi:<https://doi.org/10.1016/j.radmeas.2011.07.008> (2011).
- 10 Wei, W. *et al.* Monolithic integration of hybrid perovskite single crystals with heterogenous substrate for highly sensitive X-ray imaging. *Nat. Photonics.* **11**, 315-321 (2017).
- 11 Zhuang, R. *et al.* Highly sensitive X-ray detector made of layered perovskite-like $(\text{NH}_4)_3\text{Bi}_2\text{I}_9$ single crystal with anisotropic response. *Nat. Photonics.* **13**, 602-608 (2019).
- 12 Pan, W. *et al.* $\text{Cs}_2\text{AgBiBr}_6$ single-crystal X-ray detectors with a low detection limit. *Nat. Photonics.* **11**, 726-732 (2017).
- 13 Zhang, Y. *et al.* Nucleation-controlled growth of superior lead-free perovskite $\text{Cs}_3\text{Bi}_2\text{I}_9$ single-crystals for high-performance X-ray detection. *Nat. Commun.* **11**, 2304 (2020).
- 14 Liu, Y. *et al.* Inch-size 0D-structured lead-free perovskite single crystals for highly sensitive stable X-ray imaging. *Matter* **3**, 180-196 (2020).
- 15 Xia, M. *et al.* Unveiling the structural descriptor of $\text{A}_3\text{B}_2\text{X}_9$ perovskite derivatives toward X-ray detectors with low detection limit and high stability. *Adv. Funct. Mater.* **30**, 1910648 (2020).
- 16 Kasap, S. *et al.* Amorphous selenium and its alloys from early xeroradiography to high resolution X-ray image detectors and ultrasensitive imaging tubes. *Phys. Status Solidi B* **246**, 1794-1805 (2009).
- 17 Matt, G. J. *et al.* Sensitive direct converting X-ray detectors utilizing crystalline CsPbBr_3 perovskite films fabricated via scalable melt processing. *Adv. Mater. Interfaces* **7**, 1901575 (2020).

- 18 Xu, M. *et al.* Orientation and mobility control of 4HCB organic film for flexible X-ray detectors with high performance. *J. Mater. Sci. Technol.* **135**, 46-53, doi:<https://doi.org/10.1016/j.jmst.2022.06.045> (2023).
- 19 Jinno, H. *et al.* Self-powered ultraflexible photonic skin for continuous bio-signal detection via air-operation-stable polymer light-emitting diodes. *Nat. Commun.* **12**, 2234, doi:10.1038/s41467-021-22558-6 (2021).
- 20 Beyer, G. P. *et al.* An implantable MOSFET dosimeter for the measurement of radiation dose in tissue during cancer therapy. *IEEE Sens. J.* **8**, 38-51 (2008).
- 21 Szeles, C. Advances in the crystal growth and device fabrication technology of CdZnTe room temperature radiation detectors. *IEEE Trans. Nucl. Sci.* **51**, 1242-1249, doi:10.1109/TNS.2004.829391 (2004).
- 22 He, Z. Review of the Shockley–Ramo theorem and its application in semiconductor gamma-ray detectors. *Nucl. Instrum. Methods. Phys. Res. A* **463**, 250-267, doi:[https://doi.org/10.1016/S0168-9002\(01\)00223-6](https://doi.org/10.1016/S0168-9002(01)00223-6) (2001).
- 23 Zhao, D. *et al.* Purely organic 4HCB single crystals exhibiting high hole mobility for direct detection of ultralow-dose X-radiation. *J. Mater. Chem. A* **8**, 5217-5226, doi:10.1039/c9ta12817d (2020).
- 24 Ciavatti, A., Sellin, P. J., Basiricò, L., Fraleoni-Morgera, A. & Fraboni, B. Charged-particle spectroscopy in organic semiconducting single crystals. *Appl. Phys. Lett.* **108**, 153301, doi:10.1063/1.4945597 (2016).
- 25 Paschalis, P., Mavromichalaki, H. & Dorman, L. A quantitative study of the 6NM-64 neutron monitor by using Geant4: 1. Detection efficiency for different particles. *Nucl. Instrum. Methods. Phys. Res. A* **729**, 877-887 (2013).
- 26 Shrestha, S. *et al.* High-performance direct conversion X-ray detectors based on sintered hybrid lead triiodide perovskite wafers. *Nat. Photonics.* **11**, 436-440 (2017).
- 27 Song, Y. *et al.* Detector-grade perovskite single-crystal wafers via stress-free gel-confined solution growth targeting high-resolution ionizing radiation detection. *Light Sci. Appl.* **12**, doi:10.1038/s41377-023-01129-y (2023).
- 28 Jiang, J. *et al.* Synergistic strain engineering of perovskite single crystals for highly stable and sensitive X-ray detectors with low-bias imaging and monitoring. *Nat. Photonics.* **16**, 575-581, doi:10.1038/s41566-022-01024-9 (2022).
- 29 Pan, L. *et al.* Ultrahigh-Flux X-ray Detection by a Solution-Grown Perovskite

- CsPbBr₃ Single-Crystal Semiconductor Detector. *Adv. Mater.*, 2211840, doi:10.1002/adma.202211840 (2023).
- 30 He, Y. *et al.* Sensitivity and Detection Limit of Spectroscopic-Grade Perovskite CsPbBr₃ Crystal for Hard X-Ray Detection. *Adv. Funct. Mater.* **32**, 2112925, doi:10.1002/adfm.202112925 (2022).
- 31 Xu, Y. *et al.* Characterization of CdZnTe crystals grown using a seeded modified vertical bridgman method. *IEEE Trans. Nucl. Sci.* **56**, 2808-2813 (2009).
- 32 Wang, F. *et al.* Precursor engineering for solution method-grown spectroscopy-grade CsPbBr₃ crystals with high energy resolution. *Chem. Mater.* **34**, 3993-4000 (2022).
- 33 Zhao, D. *et al.* Direct Detection of Fast Neutrons by Organic Semiconducting Single Crystal Detectors. *Adv. Funct. Mater.* **32**, 2108857, doi:10.1002/adfm.202108857 (2022).

REVIEWERS' COMMENTS

Reviewer #2 (Remarks to the Author):

Very nice work and convincing discussions. But it is clear that the detector is still in its infancy, with real experimentally acquired alpha spectra far worse than that of CZT and Si. However, this is a biocompatible organic semiconductor detector and is worth publication.

A few more comments:

“spectra detection time down to 3 39 ms. “ “a 328 pulse-mode acquisition time of 3 ms is sufficient to acquire a resolvable spectrum”

What's the point of having a short counting time? The counting time depends on the intensity of the source. Why would people want to measure a spectrum with a short counting time? Normally, the preference is for a longer counting time to achieve better spectroscopy performance.

“pulse-mode operation at high event rates, leading to the real-320 time beam monitoring using a biocompatible”

Pulse-mode operation at high event rates doesn't lead to real-time beam monitoring using a biocompatible detector; the current mode is used for high-rate beam monitoring. High event rates can cause problems with pulse pile-up in pulse mode.

The authors claim, "In addition, the 4HPA detectors have a response for fast neutrons (Fig. S17-S18)."

Neutron detection is a much more complicated and error-prone process (for example, gamma discrimination). I suggested to the author that they should discuss this in a separate paper instead of making a single statement and referring to a figure in the supplemental material.

Reviewer #3 (Remarks to the Author):

The paper can be accepted now.

Reviewer #2 (Remarks to the Author):

Very nice work and convincing discussions. But it is clear that the detector is still in its infancy, with real experimentally acquired alpha spectra far worse than that of CZT and Si. However, this is a biocompatible organic semiconductor detector and is worth publication.

Answer: We thank for the reviewer's positive comments.

A few more comments:

1. "spectra detection time down to 3 ms." "a pulse-mode acquisition time of 3 ms is sufficient to acquire a resolvable spectrum"

What's the point of having a short counting time? The counting time depends on the intensity of the source. Why would people want to measure a spectrum with a short counting time? Normally, the preference is for a longer counting time to achieve better spectroscopy performance.

Answer: We appreciate for the reviewer's comment. Short counting time means that we can achieve a **real-time** spectra detection of charged particles with **single-particle sensitivity** and energy resolving ability by a **compact** and **biocompatible** semiconductor sensor. No previous work (both organic and inorganic detectors) can satisfy all requirements of **real-time measurement of energy deposition, single-particle sensitivity, biocompatibility, compact detector size, and low cost**, which are very important criteria for real-time in-vivo dosimeter (IVD)¹⁻³. IVD with those properties can reduce the uncertainty and error of external beam radiotherapy (**EBRT**) by measuring the dose delivered in the body with higher sensitivity and improved dose response non-linearity than conventional integrated detection mode, leading to more effective and safer closed-loop radiotherapy treatment.

2. "pulse-mode operation at high event rates, leading to the real time beam monitoring using a biocompatible"

Pulse-mode operation at high event rates doesn't lead to real-time beam monitoring using

a biocompatible detector; the current mode is used for high-rate beam monitoring. High event rates can cause problems with pulse pile-up in pulse mode.

Answer: We appreciate for the reviewer's comment. The "high event rate" here means compared with other organic detectors with very low counting rate (pulse time of around hundreds of microseconds), e.g. 4HCB detectors, 4HPA with better charge transport properties can operate at higher event rate.

We have revised in main manuscript to avoid the possible misunderstanding:

"pulse-mode operation at higher event rates compared with other organic detectors."

3. The authors claim, "In addition, the 4HPA detectors have a response for fast neutrons (Fig. S17-S18).

Neutron detection is a much more complicated and error-prone process (for example, gamma discrimination). I suggested to the author that they should discuss this in a separate paper instead of making a single statement and referring to a figure in the supplemental material.

Answer: We appreciate for the reviewer's comment. We will discuss the fast neutron detection performance in a separate paper. We gave a short discussion in this paper because the fast neutron detection by organic semiconductors and the detailed discussion have been demonstrated in our previous paper⁴. The stable and accurate detection system has been established, other issues such as neutron/gamma discrimination, long-term stability also were defined and discussed. The presented result is reliable.

References

- 1 Bloemen-van Gorp, E. J. *et al.* In Vivo Dosimetry With a Linear MOSFET Array to Evaluate the Urethra Dose During Permanent Implant Brachytherapy Using Iodine-125. *International Journal of Radiation Oncology Biology Physics* **75**, 1266-1272, doi:<https://doi.org/10.1016/j.ijrobp.2009.04.042> (2009).
- 2 Wang, L. L. W. *et al.* Determination of the quenching correction factors for plastic scintillation detectors in therapeutic high-energy proton beams. *Phys. Med. Biol.* **57**, 7767-7781, doi:10.1088/0031-9155/57/23/7767 (2012).

- 3 Lee, K. *et al.* A Millimeter-Scale Single Charged Particle Dosimeter for Cancer Radiotherapy. *IEEE Journal of Solid-State Circuits* **55**, 2947-2958, doi:10.1109/JSSC.2020.3024231 (2020).
- 4 Zhao, D. *et al.* Direct Detection of Fast Neutrons by Organic Semiconducting Single Crystal Detectors. *Adv. Funct. Mater.* **32**, 2108857, doi:10.1002/adfm.202108857 (2022).